# A Single Recipe for Online Submodular Maximization with Adversarial or Stochastic Constraints

**Omid Sadeghi**
Department of Electrical and Computer Engineering
University of Washington
Seattle, WA 98195
omids@uw.edu

**Prasanna Raut**
Department of Mechanical Engineering
University of Washington
Seattle, WA 98195
raut@uw.edu

**Maryam Fazel**
Department of Electrical and Computer Engineering
University of Washington
Seattle, WA 98195
mfazel@uw.edu

## Abstract

In this paper, we consider an online optimization problem in which the reward functions are DR-submodular, and in addition to maximizing the total reward, the sequence of decisions must satisfy some convex constraints on average. Specifically, at each round $t \in \{1, \ldots, T\}$, upon committing to an action $x_t$, a DR-submodular utility function $f_t(\cdot)$ and a convex constraint function $g_t(\cdot)$ are revealed, and the goal is to maximize the overall utility while ensuring the average of the constraint functions $\frac{1}{T} \sum_{t=1}^{T} g_t(x_t)$ is non-positive. Such cumulative constraints arise naturally in applications where the average resource consumption is required to remain below a prespecified threshold. We study this problem under an adversarial model and a stochastic model for the convex constraints, where the functions $g_t$ can vary arbitrarily or according to an i.i.d. process over time slots $t \in \{1, \ldots, T\}$, respectively. We propose a single algorithm which achieves sub-linear (with respect to $T$) regret as well as sub-linear constraint violation bounds in both settings, without prior knowledge of the regime. Prior works have studied this problem in the special case of linear constraint functions. Our results not only improve upon the existing bounds under linear cumulative constraints, but also give the first sub-linear bounds for general convex long-term constraints.

## 1 Introduction

Online optimization covers a large number of problems in which information is revealed incrementally (i.e., *online*) and irrevocable decisions should be made at each step in face of uncertainty about the future arriving information [1–5]. Such problems could be formulated as a repeated game between the decision maker (i.e., the learner) and the adversary (i.e., the nature or environment). At each iteration of this game, the learner chooses an action from a fixed domain set and then, it receives feedback in the form of utility or reward for her selected action. In the non-stochastic feedback model, no assumptions are made on the sequence of arriving rewards except their boundedness. As time goes by, the learner aims to observe the past and make better decisions to maximize the overall reward. The performance of online algorithms are usually measured through the *regret* or the *competitive ratio* of the algorithm. In the regret analysis framework, at each round, the learner has to commit to an action before observing the corresponding reward function and the goal is to design algorithms whose total accumulated reward differs sub-linearly (in the time horizon $T$) from the reward of the

best fixed benchmark action (or sequence) with hindsight information [4, 6–8]. On the other hand, in the competitive analysis setting, the decision maker is allowed to first observe the reward function at each step and then, choose her action accordingly (i.e., the *1-lookahead setting*). In this setting, the goal is to obtain bounds for the ratio of the total reward of the algorithm and the offline optimum (i.e., the competitive ratio) [9, 10]. In this work, we focus on the regret analysis setting.

In most of the prior work on online learning, there are no constraints on the sequence of decisions made by the learner and maximizing the overall reward is the sole objective [4, 5]. However, in many applications, there indeed exist some constraints on the decisions of the algorithm which need to be satisfied on average [11–14]. For instance, in an online task assignment problem in crowdsourcing markets, the requester needs to balance her total payment to workers against a prespecified allotted budget [15]. The advertiser in an online ad placement problem has a limited budget to invest on buying ads on different websites [16–18]. Note that in both of these problems, the resource (budget) consumption at each round is not known ahead of time. In crowdsourcing, the consumed resource depends on the workers' overall cost for performing the task, and even if a worker's hourly rate is known, the length of time required may not be known beforehand; in the online ad allocation problem, resource use depends on the number of clicks on the ads.

## 1.1 Related work

**Online submodular maximization.** Consider an online unconstrained optimization problem in which the reward functions are monotone DR-submodular. [19] proposed the Meta-Frank-Wolfe algorithm for this problem and obtained $\mathcal{O}(\sqrt{T})$ regret bound against the $(1 - \frac{1}{e})$ approximation to the best fixed decision in hindsight where $(1 - \frac{1}{e})$ is the best polynomial-time approximation ratio in the offline setting. The Meta-Frank-Wolfe algorithm requires access to the full gradient of the reward functions and performs $\mathcal{O}(\sqrt{T})$ gradient evaluations per step. More recently, [20] generalized this algorithm to the setting where only stochastic gradient estimates are available. Moreover, [21] proposed the Mono-Frank-Wolfe algorithm which performs only one gradient evaluation per round and requires only unbiased estimates of the gradient.

**Online optimization with adversarial constraints.** Online convex optimization with constraints, where both the convex objective functions $\{f_t\}_{t=1}^T$ and the convex constraint functions $\{g_t\}_{t=1}^T$ can vary arbitrarily, was first studied by [22]. They provided a surprisingly simple counterexample which showed that it is not always possible to achieve a sub-linear regret against the best fixed benchmark action in hindsight while the total constraint violation is sub-linear. Therefore, subsequent works added more assumptions to the problem setting to be able to obtain meaningful results. In particular, not only did they require the fixed benchmark action to satisfy the long-term constraint (i.e., $\sum_{t=1}^T g_t(x^*) \leq 0$), but they also restricted the benchmark to satisfy the constraint proportionally over any window of size $W \in [1, T]$. In other words, the fixed comparator action was required to be chosen from the set $\mathcal{X}_W = \{x \in \mathcal{X} : \sum_{\tau=t}^{t+W-1} g_\tau(x) \leq 0, \ 1 \leq t \leq T - W + 1\}$. Note that if $W = T$, we recover the usual definition of the benchmark and the smaller $W$ is, the comparator action is more restricted. See Table 1 for an overview of results under different choices of window length. Note that the setting in [17] is different, the objective functions are monotone DR-submodular (and generally non-concave) and the constraint functions are linear.

| Paper | Cost (utility) | Constraint | Window size | Regret | Constraint violation |
|-------|---------------|-----------|-------------|--------|---------------------|
| [23] | convex | convex | 1 | $\mathcal{O}(\sqrt{T})$ | $\mathcal{O}(T^{\frac{3}{4}})$ |
| [24] | convex | convex | 1 | $\mathcal{O}(\sqrt{T})$ | $\mathcal{O}(\sqrt{T})$ |
| [16][a] | convex | convex | $W$ | $\mathcal{O}(\sqrt{T} + \frac{WT}{V})$ | $\mathcal{O}(\sqrt{VT})$ |
| [17] | DR-submodular | linear | $W$ | $\mathcal{O}(\sqrt{WT})$ | $\mathcal{O}(W^{\frac{1}{4}}T^{\frac{3}{4}})$ |

Table 1: Prior results for online problems with adversarial cumulative constraints in various settings. Note that in (a), $V \in (W, T)$ is a tunable parameter.

The performance of online convex optimization with adversarial convex constraints has also been analyzed against a dynamic benchmark sequence (i.e., *dynamic regret*) and sub-linear regret and constraint violation bounds have been derived under full and bandit feedback settings [25, 26].

**Online optimization with stochastic constraints.** In light of the aforementioned impossibility result of [22], many subsequent works focused on stochastically time-varying constraints in which the constraint functions over $t \in [T]$ are assumed to be an i.i.d. process. In this setting, the benchmark

action is required to satisfy the constraint in expectation, i.e., $\mathbb{E}[g_t(x^*)] \leq 0 \ \forall t \in [T]$. In this framework, [24, 27, 28] obtained $\mathcal{O}(\sqrt{T})$ regret and constraint violation bounds simultaneously, both in expectation and with high probability. Outside of the convex setting, [18] analyzed this problem for monotone DR-submodular utility functions and linear constraint functions. They managed to obtain $\mathcal{O}(\sqrt{T})$ constraint violation bound in expectation and with high probability. In addition, they derived $\mathcal{O}(T^{\frac{3}{4}})$ and $\mathcal{O}(\sqrt{T})$ regret bounds, in expectation and with high probability respectively.

## 1.2 Contributions

In this paper, we focus on a general class of online optimization problems where the reward functions $\{f_t\}_{t=1}^T$ are monotone DR-submodular and are chosen adversarially. Moreover, the constraint functions $\{g_t\}_{t=1}^T$ are monotone and convex. We study this problem in two settings. In the first setting, the constraint functions are assumed to vary arbitrarily. In the second model, we further restrict the sequence of constraint functions to be an i.i.d. process over time slots $t \in [T]$. We make the following contributions:

• Inspired by the Meta-Frank-Wolfe algorithm of [19] and the algorithm of [24], we propose Algorithm 1 in Section 4 for both adversarial or stochastic constraints without prior knowledge of the regime. In particular, for the adversarial setting, we obtain an $\mathcal{O}(T^{1-\frac{\epsilon}{2}})$ static regret bound against the benchmark with window length $W = T^{1-\epsilon}$, and an $\mathcal{O}(T^{1-\frac{\epsilon}{2}})$ total constraint violation bound. Moreover, if we consider dynamic regret as the utility performance metric, we obtain $\mathcal{O}(\sqrt{TP_T^*})$ bounds for both the dynamic regret and the total constraint violation where $P_T^* := \sum_{t=1}^{T-1} \|x_{t-1}^* - x_t^*\|$. In the setting with stochastic constraints, using the same algorithm (Algorithm 1), we obtain $\mathcal{O}(\sqrt{T})$ regret and total constraint violation bounds, both in expectation and with high probability.

• In Section 5, we propose Algorithm 2 which is based on the Mono-Frank-Wolfe algorithm of [21]. Compared to Algorithm 1, Algorithm 2 is computationally more efficient, but it achieves slightly worse performance guarantees. In particular, Algorithm 2 obtains an $\mathcal{O}(T^{\frac{2}{3}})$ static regret against the benchmark with window size $W \in [1, T^{\frac{1}{3}}]$ and an $\mathcal{O}(T^{\frac{2}{3}})$ total constraint violation bound. Similar bounds can also be derived in the stochastic setting, both in expectation and with high probability.

Lastly, we validate our theoretical findings and demonstrate the advantages of our proposed algorithms over prior work in a series of numerical experiments in Section 6.

Proofs for all the claims and results in the paper are provided in the appendix.

## 1.3 Notations

We use $[T]$ to denote the set $\{1, 2, \dots, T\}$. For a vector $x \in \mathbb{R}^n$, we use $x_i$ to denote the $i$-th entry of $x$. For $u \in \mathbb{R}$, the notation $[u]_+$ denotes the application of function $\max\{x, 0\}$, i.e., $[u]_+ = \max\{u, 0\}$. The inner product of two vectors $x, y \in \mathbb{R}^n$ is denoted by either $\langle x, y \rangle$ or $x^T y$. Also, for two vectors $x, y \in \mathbb{R}^n$, we write $x \preceq y$ if $x_i \leq y_i$ holds for every $i \in [n]$. A function $f : \mathbb{R}^n \to \mathbb{R}$ is called monotone if for all $x, y$ such that $x \preceq y$, $f(x) \leq f(y)$ holds. For a vector $x \in \mathbb{R}^n$, we use $\|x\|$ to denote the Euclidean norm of $x$. A differentiable function $f : \mathcal{X} \to \mathbb{R}$ is $\beta$-Lipschitz if for all $x, y \in \mathcal{X}$, we have $|f(y) - f(x)| \leq \beta \|y - x\|$, or equivalently, $\|\nabla f(x)\| \leq \beta$ holds. For a convex set $\mathcal{X}$, we will use $\mathcal{P}_{\mathcal{X}}(y) = \arg\min_{x \in \mathcal{X}} \|x - y\|$ to denote the projection of $y$ onto set $\mathcal{X}$. The diameter of a set $\mathcal{X}$ is defined as $\max_{x, y \in \mathcal{X}} \|y - x\|$.

## 2 DR-submodular functions

We say that a differentiable function $f : \mathcal{X} \to \mathbb{R}$, $\mathcal{X} \subset \mathbb{R}_+^n$, is DR-submodular if its gradient is an order-reversing mapping, i.e., we have:

$$x \succeq y \Rightarrow \nabla f(x) \preceq \nabla f(y).$$

For a twice differentiable function $f$, it is DR-submodular if and only if its Hessian matrix $\nabla^2 f$ is entry-wise non-positive. It is noteworthy that although DR-submodularity and concavity are equivalent for the special case of $n = 1$, DR-submodular functions are generally non-concave. Nonetheless, an important consequence of DR-submodularity is concavity along non-negative directions [29, 30], i.e., for all $x, y$ such that $x \preceq y$, we have $f(y) \leq f(x) + \langle \nabla f(x), y - x \rangle$.

For a DR-submodular function $f$, we say that $f$ is $L$-smooth over non-negative directions if:

$$f(y) - f(x) \geq \langle \nabla f(x), y - x \rangle - \frac{L}{2} \|y - x\|^2 \ \ \forall x, y; x \preceq y.$$

There are many functions which satisfy the DR-submodularity property. In particular, we mention a number of them in Appendix A which have been used for the experiments in Section 6.

## 3 Problem formulation

We consider the following protocol for online DR-submodular maximization with long-term convex constraints. At each iteration $t \in [T]$, the online algorithm chooses an action $x_t \in \mathcal{X}$ where $\mathcal{X}$ is the fixed domain set. Upon committing to this action, (i) a monotone DR-submodular utility function $f_t : \mathcal{X} \to \mathbb{R}$ is revealed and the algorithm obtains the reward $f_t(x_t)$ and (ii) a monotone convex constraint function $h_t : \mathcal{X} \to \mathbb{R}$ is revealed and $h_t(x_t)$ amount of resource is consumed. The total available resource is denoted by $B_T$ which is given offline. Also, we assume the horizon $T$ is known in advance, however, if $T$ is not available offline, we can use the well-known doubling trick to obtain the same performance guarantees with slightly worse constants. The goal is to maximize the overall obtained reward while ensuring the resource constraint is satisfied on average, i.e., $\lim_{T \to \infty} \frac{1}{T} \left( \sum_{t=1}^{T} h_t(x_t) - B_T \right) \leq 0$. Denoting $g_t(\cdot) = h_t(\cdot) - \frac{B_T}{T}$, the resource constraint could be written as $\lim_{T \to \infty} \frac{1}{T} \sum_{t=1}^{T} g_t(x_t) \leq 0$. In other words, we aim to maximize the overall utility while ensuring the total constraint violation $\sum_{t=1}^{T} g_t(x_t)$ grows sub-linearly in $T$. The offline optimization problem is as follows:

$$
\begin{aligned}
\text{maximize} \quad & \textstyle\sum_{t=1}^{T} f_t(x_t) \\
\text{subject to} \quad & \textstyle\sum_{t=1}^{T} g_t(x_t) \leq 0 \\
& x_t \in \mathcal{X} \; \forall t \in [T].
\end{aligned}
\tag{1}
$$

We study this problem under two settings. In the first online model, we assume that for all $t \in [T]$, both the utility function $f_t$ and the constraint function $h_t$ are chosen adversarially. In other words, we do not make any assumptions on the arriving functions $f_t$ and $h_t$. In the second model, while the utility function $f_t$ is still assumed to be arbitrary, the constraint function $h_t$ is a random i.i.d. sample drawn from some unknown underlying distribution over a class $\mathcal{H}$ of monotone convex functions.
Note that our proposed algorithms can easily handle multiple online convex constraints with the same performance guarantees. However, for ease of notation, we focus on the special case of a single resource constraint.
To analyze this online optimization problem, we will make a number of assumptions that are common for online submodular problems.
**Assumption 1.** $\mathcal{X}$ is a convex and compact set with diameter $R$, and $0 \in \mathcal{X}$.
**Assumption 2.** For all $t \in [T]$, the reward function $f_t$ is monotone, DR-submodular, $\beta_f$-Lipschitz, $L$-smooth along non-negative directions and normalized (i.e., $f_t(0) = 0$).
**Assumption 3.** For all $t \in [T]$, the constraint function $h_t$ is monotone, convex, $\beta_h$-Lipschitz and normalized (i.e., $h_t(0) = 0$). In the stochastic setting, we assume that these assumptions hold for all $h \in \mathcal{H}$. Note that since $g_t$ was defined as $g_t(\cdot) = h_t(\cdot) - \frac{B_T}{T}$, these assumptions apply to $g_t$ as well. Since $\mathcal{X}$ is compact and $f_t, h_t \; \forall t \in [T]$ are $\beta$-Lipschitz, where $\beta = \max\{\beta_f, \beta_h\}$, $f_t(\cdot)$ and $g_t(\cdot) = h_t(\cdot) - \frac{B_T}{T}$ are both bounded, i.e., $|f_t(x)| \leq F$ and $|g_t(x)| \leq G$ for all $x \in \mathcal{X}$ and $t \in [T]$.

### 3.1 Motivating Applications

There are a number of interesting applications that could be cast in our framework. In particular, we have described three examples below which have been used in Section 6 for the numerical experiments. See [17, 18] for more examples. In all the following applications, if the utility function is a submodular set function, we apply our algorithms to the DR-submodular continuous extension of the set function, and use the lossless pipage rounding technique of [30] to make integral allocations.
**Online joke recommendation.** In this problem, we aim to design a joke recommendation algorithm to assign jokes to a sequence of users arriving online such that the overall impression of the jokes are maximized in a fixed time horizon $B_T$. At each step $t \in [T]$, a user arrives and the algorithm should assign a bundle of at most $m$ jokes, $x_t \in \{x \in \{0,1\}^n : 1^T x \leq m\}$, to her. If joke $i$ is assigned to user $t$, she spends $[p_t]_i$ amount of time to read the joke and submit her rating $[r_t]_i$. In other words, $h_t(x) = \langle p_t, x \rangle$. The overall impression is the submodular set function $F_t(x) = r_t^T x + \sum_{i,j:i<j} \theta_{ij}^{(t)} x_i x_j$ where $\theta_{ij}^{(t)} \leq 0$ penalizes the similarity of jokes $i$ and $j$. This function has been extensively used in the literature to encourage diversity [31].
**Online task assignment in crowdsourcing markets.** In this problem, there exists a requester with

a limited budget $B_T$ that submit jobs and benefits from them being completed. There are $n$ types of jobs available to be assigned to workers arriving online. At each step $t \in [T]$, a worker arrives and the requester has to assign a bundle $x_t \in \mathcal{X} = \{x \in \mathbb{R}_+^n : 0 \preceq x \preceq 1\}$ of the jobs to the worker. The worker has a private cost $[p_t]_i \ \forall i \in [n]$ for performing one unit of the assigned job $i$, where $[p_t]_i$ denotes the $i$-th entry of vector $p_t$. In other words, we have $h_t(x) = \langle p_t, x \rangle$. The rewards obtained by the requester from this job assignment is a DR-submodular function $f_t(x) = \sum_{i=1}^n [u_t]_i \log(1 + x_i) + \sum_{i,j:i \neq j} [\theta_t]_{ij} x_i x_j$, where $[u_t]_i \geq 0$ and $[\theta_t]_{ij} \leq 0$. The DR-submodularity of the utility function captures the diminishing returns of assigning more jobs to the worker, i.e., as the number of assigned jobs to the worker increases, she has less time, energy and resource available to devote to each fixed job $i \in [n]$ and therefore, the reward (quality of the completed task) obtained from the worker performing one unit of job $i$ decreases. In other words, if $x \preceq y$, $\nabla_i f(x) \geq \nabla_i f(y) \ \forall i \in [n]$ holds. The goal is to maximize the overall reward obtained by the requester while the budget constraint is satisfied on average.

**Online welfare maximization with production cost.** In this problem, there is a seller who has $n$ types of products for sale that may be produced on demand using a fixed limited budget $B_T$. At each step $t \in [T]$, an agent (customer) arrives online and the seller has to assign a bundle $x_t \in \mathcal{X} = \{x \in \mathbb{R}_+^n : 0 \preceq x \preceq 1\}$ of products to the agent. The production cost for this assignment is $h_t(x_t) = x_t^T P_t x_t$, where $P_t$ is an entry-wise non-negative positive definite matrix. Quadratic production cost functions with increasing gradient are commonly used in the literature [32, 33]. The agent has an unknown private DR-submodular valuation function $f_t(x) = \log \det \big( \text{diag}(x)(L_t - I) + I \big)$ over the items, where $L_t$ is a positive semidefinite matrix and the DR-submodularity property characterizes the diversity of the assigned bundle. Therefore, the utility obtained by assigning the bundle $x_t$ equals $f_t(x_t)$. The goal is to maximize the overall valuation of the agents while satisfying the budget constraint of the seller on average.

## 3.2 Benchmarks

We measure the performance of our proposed algorithms with the notions of *regret* and *total constraint violation* to quantify the overall utility and total resource consumption of the algorithms respectively. We define these notions below.

**Total constraint violation**. The total constraint violation of an online algorithm with outputs $\{x_t\}_{t=1}^T$ is the following:

$$C_T := \sum_{t=1}^T g_t(x_t) = \sum_{t=1}^T h_t(x_t) - B_T.$$

We aim to design algorithms whose total constraint violation is sub-linear in $T$.

In Online Convex Optimization (OCO), the utility performance of the algorithm is usually compared against *static* or *dynamic* benchmark sequences. Static regret metric has been extensively used in the literature [7,8,34–37]. However, in problems where the environment is changing (i.e., dynamic), static regret is no longer a suitable measure and an enhanced measure, i.e., dynamic regret, is used [38–42]. In our setting, the regret metric should also specify how the benchmark actions behave with respect to the long-term adversarial or stochastic constraint. We thus introduce three notions of regret below.

**Adversarial regret**. In the adversarial setting, where both the utility and constraint functions are chosen arbitrarily, the $(1 - \frac{1}{e})$-regret of an algorithm with outputs $\{x_t\}_{t=1}^T$ against a static benchmark action $x_W^*$ with window length $W \in [1, T]$ is defined as:

$$R_{W,T}^{(A,S)} = (1 - \frac{1}{e}) \sum_{t=1}^T f_t(x_W^*) - \sum_{t=1}^T f_t(x_t),$$

where:

$$x_W^* = \arg\max_{x \in \mathcal{X}_W} \sum_{t=1}^T f_t(x), \ \mathcal{X}_W = \{x \in \mathcal{X} : \sum_{\tau=t}^{t+W-1} g_\tau(x) \leq 0, \ 1 \leq t \leq T - W + 1\}.$$

Furthermore, in this adversarial setting, the $(1 - \frac{1}{e})$-regret against a dynamic benchmark sequence $\{x_t^*\}_{t=1}^T$ is as follows:

$$R_T^{(A,D)} = (1 - \frac{1}{e}) \sum_{t=1}^T f_t(x_t^*) - \sum_{t=1}^T f_t(x_t),$$

**Algorithm 1**

**Input:** $\mathcal{X}$ is the constraint set, $T$ is the horizon, $V > 0$, $\alpha > 0$ and $K \in \mathbb{N}$.
**Output:** $\{x_t : 1 \leq t \leq T\}$.
Initialize $\lambda_1^{(k)} = v_0^{(k)} = x_0^{(k)} = 0 \; \forall k \in [K]$.
**for** $t = 1$ **to** $T$ **do**
    $x_t^{(1)} = 0$.
    **for** $k = 1$ **to** $K$ **do**
        $v_t^{(k)} = \arg\max_{x \in \mathcal{X}} \left( \langle V \nabla f_{t-1}(x_{t-1}^{(k)}) - \lambda_t^{(k)} \nabla g_{t-1}(v_{t-1}^{(k)}), x \rangle - \alpha \|x - v_{t-1}^{(k)}\|^2 \right)$,
        $x_t^{(k+1)} = x_t^{(k)} + \frac{1}{K} v_t^{(k)}$.
    **end for**
    Set $x_t = x_t^{(K+1)}$ and play $x_t$.
    Observe the utility function $f_t$ and the constraint function $g_t$.
    **for** $k = 1$ **to** $K$ **do**
        $\lambda_{t+1}^{(k)} = [\lambda_t^{(k)} + g_{t-1}(v_{t-1}^{(k)}) + \langle \nabla g_{t-1}(v_{t-1}^{(k)}), v_t^{(k)} - v_{t-1}^{(k)} \rangle]_+$.
    **end for**
**end for**

where $\{x_t^*\}_{t=1}^T$ is any benchmark sequence for which $g_t(x_t^*) \leq 0$ holds.
**Stochastic regret**. In the stochastic input model, where the utility functions $\{f_t\}_{t=1}^T$ are chosen adversarially and the constraint functions $\{h_t\}_{t=1}^T$ are drawn i.i.d. according to an unknown underlying distribution over the class $\mathcal{H}$, the $(1 - \frac{1}{e})$-regret of an algorithm with outputs $\{x_t\}_{t=1}^T$ against a static benchmark action $x^*$ is the following:

$$R_T^{(S,S)} = (1 - \frac{1}{e}) \sum_{t=1}^T f_t(x^*) - \sum_{t=1}^T f_t(x_t), \; x^* = \arg\max_{x \in \mathcal{X}: \mathbb{E}[g_t(x)] \leq 0 \; \forall t \in [T]} \sum_{t=1}^T f_t(x).$$

## 4 One practical algorithm for adversarial or stochastic constraints

In this section, we propose our first algorithm that could be applied to online DR-submodular maximization problems with both adversarial or stochastic constraints without prior information about the regime. The algorithm is provided in Algorithm 1. The algorithm generalizes that of [24] for the convex setting to handle generally non-concave DR-submodular utility functions. In particular, inspired by the Meta-Frank-Wolfe algorithm of [19], we have changed the primal update of the algorithm of [24] to be able to obtain regret bounds in this setting. The following lemma provides an equivalent formulation of the primal update of the algorithm.

**Lemma 1.** *For all $t \in [T]$ and $k \in [K]$, the update rule of Algorithm 1 for $v_t^{(k)}$ is equivalent to the following:*

$$v_t^{(k)} = \mathcal{P}_{\mathcal{X}} \left( v_{t-1}^{(k)} + \frac{1}{2\alpha} \left( V \nabla f_{t-1}(x_{t-1}^{(k)}) - \lambda_t^{(k)} \nabla g_{t-1}(v_{t-1}^{(k)}) \right) \right),$$

*where $\mathcal{P}_{\mathcal{X}}$ denotes the projection onto set $\mathcal{X}$.*

Thus, for each $k \in [K]$, the algorithm runs an instance of online gradient ascent with step size $\frac{1}{2\alpha}$ to choose the point $v_t^{(k)}$, $\forall t \in [T]$, and upon committing to this action, it receives a reward of $\langle V \nabla f_t(x_t^{(k)}), v_t^{(k)} \rangle - \lambda_{t+1}^{(k)} g_t(v_t^{(k)})$. Note that using the average of the output of $K$ online maximization algorithms to obtain $\{x_t\}_{t=1}^T$ is common in online submodular maximization [19–21]. The parameter $V$ characterizes the trade-off between maximizing the reward and satisfying the resource constraint. In other words, choosing a larger $V$ leads to higher overall reward while the constraint is further violated. The output of the algorithm at each step, $x_t$, $\forall t \in [T]$, is the average of $K$ vectors $v_t^{(k)}$ in the convex domain $\mathcal{X}$; hence, $x_t \in \mathcal{X}$ also holds.

Furthermore, the algorithm needs to maintain $K$ dual variables at every time step $t$, $\lambda_t^{(k)}$ $k \in [K]$ (as opposed to a single dual variable in [24]). To better understand the algorithm, we first provide the following two lemmas.

**Lemma 2.** *The cumulative constraint violation of Algorithm 1 could be bounded as follows:*

$$C_T \leq \frac{1}{K} \sum_{k=1}^K \lambda_{T+1}^{(k)} + \frac{\beta^2 T}{4V} + \frac{V}{K} \sum_{k=1}^K \sum_{t=1}^T \|v_t^{(k)} - v_{t-1}^{(k)}\|^2.$$

**Lemma 3.** *Let* $\Delta_t^{(k)} := \frac{(\lambda_{t+1}^{(k)})^2}{2} - \frac{(\lambda_t^{(k)})^2}{2}$ *for all* $t \in [T]$ *and* $k \in [k]$. *We have:*

$$\Delta_t^{(k)} \leq \frac{(G + \beta R)^2}{2} + \lambda_t^{(k)}\big(g_{t-1}(v_{t-1}^{(k)}) + \langle \nabla g_{t-1}(v_{t-1}^{(k)}), v_t^{(k)} - v_{t-1}^{(k)}\rangle\big).$$

$\Delta_t^{(k)}$ has been commonly used in the literature and is called Lyapunov quadratic drift [43]. The result of Lemma 2 suggests that in order to minimize the total constraint violation $C_T$, the algorithm needs to maintain small dual variables. Equivalently, for all $t \in [T]$ and $k \in [K]$, the drift of the dual variable , $\Delta_t^{(k)}$, needs to be minimized. Using the result of Lemma 3, we obtain:

$$\Delta_t^{(k)} - \langle V\nabla f_{t-1}(x_{t-1}^{(k)}), v_t^{(k)} - v_{t-1}^{(k)}\rangle + \alpha\|v_t^{(k)} - v_{t-1}^{(k)}\|^2$$

$$\leq \underbrace{\frac{(G + \beta R)^2}{2} - \langle V\nabla f_{t-1}(x_{t-1}^{(k)}) - \lambda_t^{(k)}\nabla g_{t-1}(v_{t-1}^{(k)}), v_t^{(k)} - v_{t-1}^{(k)}\rangle + \lambda_t^{(k)} g_{t-1}(v_{t-1}^{(k)}) + \alpha\|v_t^{(k)} - v_{t-1}^{(k)}\|^2}_{(a)}.$$

In Algorithm 1, $v_t^{(k)}$ is chosen to be the minimizer of (a) over $\mathcal{X}$ and the update rule for $\lambda_{t+1}^{(k)}$ corresponds to moving along the direction of the gradient of (a) with respect to the dual variable.

## 4.1 Performance guarantees

In this section, we provide the total constraint violation and regret bounds under different settings.

**Theorem 1.** *(**Total constraint violation bound**) The total constraint violation of Algorithm 1 is bounded as follows:*

$$C_T \leq \theta V + \frac{\beta^2 T}{4V} + \frac{\beta^2(1+\theta)^2 V^3 T}{4\alpha^2},$$

*where* $\theta = \max\{G + \beta R, \frac{\frac{(G+\beta R)^2}{2} + (\beta R + \frac{V\beta^2}{4\alpha})V}{VB_T/T} + \frac{\alpha R^2}{V(V+1)B_T/T} + \frac{(G+\beta R)(V+2)}{2V}\}$. *In particular, if* $\alpha \leq \mathcal{O}(V^2)$, *we have* $\theta = \mathcal{O}(1)$.

Theorem 1 characterizes the total constraint violation bound of Algorithm 1 in both adversarial and stochastic settings.

**Theorem 2.** *(**Adversarial static regret bound**) The regret of Algorithm 1 in the adversarial setting against a benchmark with window length $W$ is bounded as:*

$$R_{W,T}^{(A,S)} \leq F(W-1) + \frac{1}{2V}\min\{\theta^2 V^2, (G+\beta R)^2 \frac{(W-1)(2W-1)}{6}\} + \frac{V\beta^2(T-W+1)}{4\alpha}$$

$$+ \frac{(G+\beta R)^2(T-W+1)}{2V} + \frac{LR^2(T-W+1)}{2K} + \frac{(G+\beta R)^2(W-1)(T-W+1)}{2V} + \frac{\alpha R^2}{V}.$$

Therefore, setting $\alpha = V\sqrt{T}$, the adversarial static regret bound of Algorithm 1 is $\mathcal{O}(\frac{WT}{V} + \sqrt{T} + \frac{T}{K})$. In particular, for the adversarial setting with window size $W = T^{1-\epsilon}$, if we choose $V = \mathcal{O}(T^{1-\frac{\epsilon}{2}})$, $\alpha = V\sqrt{T}$ and $K = \mathcal{O}(T^{\frac{\epsilon}{2}})$ in Theorem 1 and Theorem 2, we have $R_{W,T}^{(A,S)} \leq \mathcal{O}(T^{1-\frac{\epsilon}{2}})$ and $C_T \leq \mathcal{O}(T^{1-\frac{\epsilon}{2}})$. In comparison, [17] obtains a similar $\mathcal{O}(T^{1-\frac{\epsilon}{2}})$ regret bound and a worse $\mathcal{O}(T^{1-\frac{\epsilon}{4}})$ total constraint violation bound, and only for the special case of linear constraint functions.

**Theorem 3.** *(**Adversarial dynamic regret bound**) The adversarial regret of Algorithm 1 against a dynamic benchmark sequence $\{x_t^*\}_{t=1}^T$ is bounded as follows:*

$$R_T^{(A,D)} \leq \frac{V\beta^2 T}{4\alpha} + \frac{(G+\beta R)^2 T}{2V} + \frac{\alpha R^2}{V} + \frac{LR^2 T}{2K} + \frac{2\alpha R P_T^*}{V},$$

*where* $P_T^* := \sum_{t=1}^{T-1}\|x_{t-1}^* - x_t^*\|$.

If we set $V = K = \mathcal{O}(\sqrt{\frac{T}{P_T^*}})$ and $\alpha = V^2$, we have $R_T^{(A,D)} \leq \mathcal{O}(\sqrt{TP_T^*})$ and $C_T \leq \mathcal{O}(\sqrt{TP_T^*})$. However, since $P_T^*$ is not known ahead of time, the parameters of the algorithm cannot be chosen as mentioned. To remedy this issue, we can extend the adaptive algorithm of [39] to our framework to obtain $\mathcal{O}(\sqrt{TP_T^*})$ regret and total constraint violation bounds simultaneously without prior knowledge of $P_T^*$.

**Theorem 4.** *(Expected Regret Bound) In the stochastic setting, the expected regret of Algorithm 1 could be bounded as follows:*

$$\mathbb{E}[R_T^{(S,S)}] \leq \frac{V\beta^2 T}{4\alpha} + \frac{(G+\beta R)^2 T}{2V} + \frac{\alpha R^2}{V} + \frac{LR^2 T}{2K}.$$

**Theorem 5.** *(High Probability Regret Bound) The regret of Algorithm 1 satisfies the following with probability at least $1-\delta$ in the stochastic setting:*

$$R_T^{(S,S)} \leq \theta G\sqrt{2T\log(\frac{1}{\delta})} + \frac{V\beta^2 T}{4\alpha} + \frac{(G+\beta R)^2 T}{2V} + \frac{\alpha R^2}{V} + \frac{LR^2 T}{2K}.$$

If we choose $V = \mathcal{O}(\sqrt{T})$, $\alpha = V^2$ and $K = \mathcal{O}(\sqrt{T})$ in Theorem 1, Theorem 4 and Theorem 5, we have $\mathbb{E}[R_T^{(S,S)}] \leq \mathcal{O}(\sqrt{T})$, $R_T^{(S,S)} \leq \tilde{\mathcal{O}}(\sqrt{T})$ w.h.p. and $C_T \leq \mathcal{O}(\sqrt{T})$. In comparison, for the special case of linear constraint functions, [18] obtains similar $\mathcal{O}(\sqrt{T})$ bounds for the total constraint violation, both in expectation and with high probability. However, despite achieving $\mathcal{O}(\sqrt{T})$ high probability regret bound, their algorithm obtains a worse $\mathcal{O}(T^{\frac{3}{4}})$ regret bound in expectation.

## 5 Trading performance for efficiency: A faster algorithm

In this section, we propose our second algorithm, presented in Algorithm 2 in the appendix, for online DR-submodular maximization problems with adversarial or stochastic constraints. Inspired by the Mono-Frank-Wolfe algorithm of [21], we divide the upcoming online rounds $1, \ldots, T$ to $Q$ equisized blocks of length $K$ (i.e., $T = QK$) and for all the rounds $t \in \{(q-1)K+1, \ldots, qK\}$ in the block $q \in Q$, we play the same action $x_q$. Using this technique, the computational complexity of Algorithm 2 reduces with a factor of $K$ compared to Algorithm 1. However, efficiency of Algorithm 2 comes at the price of slightly worse regret and total constraint violation bounds, as presented below.

**Theorem 6.** *The adversarial (static) regret bound of Algorithm 2 against the benchmark with window length $W \in [1, T^{\frac{1}{3}}]$ is as follows:*

$$\mathbb{E}[R_{W,T}^{(A,S)}] \leq \frac{V\beta^2 QK}{4\alpha} + \frac{(G+\beta R)^2 QK}{2V} + \frac{\alpha R^2 K}{V} + \frac{LR^2 Q}{2},$$

*where expectation is taken with respect to randomness of the algorithm.*

**Theorem 7.** *The total constraint violation of Algorithm 2 is bounded as below:*

$$\mathbb{E}[C_T] \leq \theta KV + \frac{\beta^2 T}{4V} + \frac{\beta^2 (1+\theta)^2 V^3 T}{4\alpha^2},$$

*where expectation is taken with respect to randomness of the algorithm.*

Thus, in the adversarial setting with window length $W \in [1, T^{\frac{1}{3}}]$, if we choose $V = K = \mathcal{O}(T^{\frac{1}{3}})$ and $\alpha = V^2$ in Theorem 6 and Theorem 7, we have $\mathbb{E}[R_{W,T}^{(A,S)}] \leq \mathcal{O}(T^{\frac{2}{3}})$ and $\mathbb{E}[C_T] \leq \mathcal{O}(T^{\frac{2}{3}})$. Note that similar to the analysis in Section 4, we can extend the above results to a benchmark with general window length in the adversarial setting. Moreover, we can obtain similar $\mathcal{O}(T^{\frac{2}{3}})$ regret and total constraint violation bounds in the stochastic setting.

## 6 Experiments

In order to verify our theoretical findings, we run our algorithms for the three experiments described in Section 3.1 and we plot the performance in Figure 1.

1) *Online joke recommendation.* We choose $n = 100$ jokes, $T = 10000$ and $B_T = 1.5T$. We vary the window length $W$ and choose $V$, $\alpha$ and $K$ according to Section 4. We set $\mathcal{X} = \{x \in [0,1]^n : 1^T x \leq 15\}$. We consider the utility functions $f_t(x) = r_t^T x + \sum_{i,j:i<j} \theta_{ij}^{(t)} x_i x_j \ \forall t \in [T]$ where $0 \leq [r_t]_i \leq 10$ is the rating of user $t$ for joke $i$ in the *Jester* dataset[1], and $\theta_{ij}^{(t)}$ is uniformly chosen from $[-0.5, 0]$. Also, $[p_t]_i$ is chosen uniformly from the range $[0.3, 6]$. We compare the overall utility

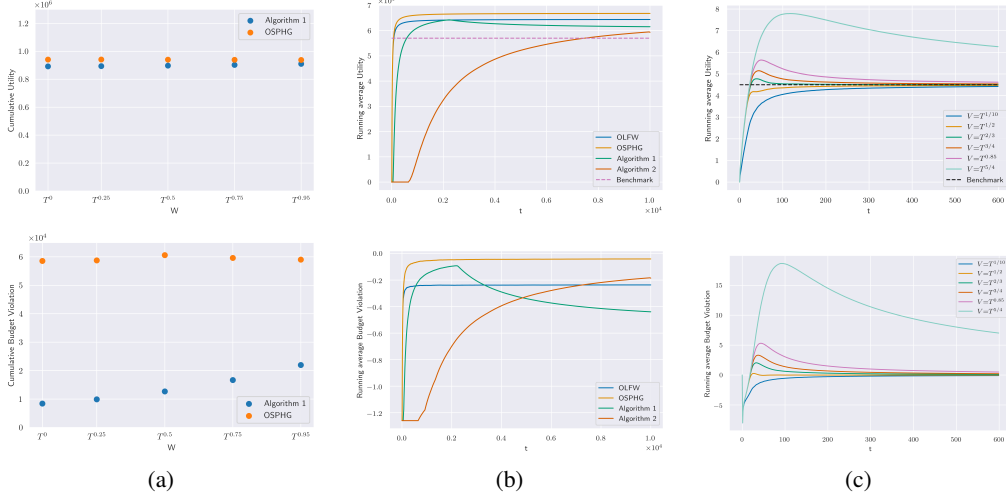

Figure 1: (a) Cumulative utility $\sum_{t=1}^{T} f_t(x_t)$ and total budget violation $\sum_{t=1}^{T} g_t(x_t)$ for experiment 1. (b) and (c) Running average of cumulative utility $\frac{1}{t}\sum_{\tau=1}^{t} f_\tau(x_\tau)$ and budget violation $\frac{1}{t}\sum_{\tau=1}^{t} g_\tau(x_\tau)$ for experiments 2 and 3 respectively.

and total budget violation of Algorithm 1 and the OSPHG algorithm of [17] for different choices of $W$. Note that we use the pipage rounding technique of [30] for both algorithms to make integral allocation of jokes to the users. Figure 1(a) verifies the superiority of Algorithm 1 compared to the OSPHG algorithm in terms of budget consumption while obtaining similar overall utility.

2) *Online task assignment in crowdsourcing markets.* We set $n = 13$, $T = 10000$ and $B_T = 0.86T$. We choose $V$, $\alpha$ and $K$ according to Section 4. We set $\mathcal{X} = \{x : 0 \preceq x \preceq 1\}$. We consider the utility functions $f_t(x) = \sum_{i=1}^{n} [u_t]_i \log(1 + x_i) + \sum_{i,j:i<j} \theta_{ij}^{(t)} x_i x_j \ \forall t \in [T]$ where $[u_t]_i$ and $\theta_{ij}^{(t)}$ are uniformly chosen from $[1, 13]$ and $[-0.07, 0]$ respectively. $[p_t]_i$ is uniformly chosen from $[0.05, 1]$. We compare the average performance of Algorithm 1, Algorithm 2, the OLFW algorithm of [18] and the OSPHG algorithm of [17]. Figure 1(b) demonstrates that Algorithm 1 strikes the right balance between the utility and budget used.

3) *Online welfare maximization with production cost.* For this experiment, we use the utility function $f_t(x) = \log \det \big( \mathrm{diag}(x)(L_t - I) + I \big)$ and the quadratic convex constraint function $h_t(x) = x^T P_t x$ for all $t \in [T]$, where $L_t$ and $P_t$ are positive definite matrices whose eigenvalues are uniformly chosen from $[2, 3]$ and $[0.3, 6]$ respectively. We consider the domain $\mathcal{X} = \{x : 0 \preceq x \preceq 1\}$. We set $n = 10$, $T = 1000$, $K = W = \sqrt{T}$ and $B_T = 4T$. We vary $V$ and choose $\alpha = V\sqrt{T}$ to see the effect of the choice of $V$ in the performance of Algorithm 1. Considering the $\mathcal{O}(\frac{WT}{V} + \sqrt{T} + \frac{T}{K})$ regret bound and $\mathcal{O}(V + \frac{T}{V})$ total constraint violation bound derived in Section 4, Figure 1(c) verifies our theoretical analysis that we need to choose $V \in (W, T)$ to obtain sub-linear regret and total constraint violation bounds simultaneously.

# 7 Conclusion

We studied an online optimization problem in which the reward functions are monotone DR-submodular, and in addition, the sequence of decisions of the learner should satisfy some adversarially or stochastically varying monotone convex constraints on average. We propose a single algorithm for both adversarial or stochastic constraints without prior knowledge of the regime. In the special case of linear constraint functions, our proposed algorithm obtains improved regret and constraint violation bounds in both adversarial and stochastic settings compared to prior work. Moreover, we derive the first sub-linear bounds for the more general case of convex constraint functions.

## Broader Impact

This theoretical paper studies online, sequential decision making with rewards and limited resources/budgets, with broad applications. The general idea of our algorithms is to be conservative enough in their resource use to guard against future unknowns, yet not miss too many opportunities over time, and to allocate limited resources better. There are many online resource allocation problems that could be cast in our framework (see Section 3.1), however, we believe that this work does not raise any potential ethical concerns.

## Acknowledgments and Disclosure of Funding

This work was supported in part by the following grants: NSF TRIPODS grant 1740551, DARPA Lagrange grant FA8650-18-2-7836, ONR MURI grant N0014-16-1-2710.

## Footnotes

[1]http://eigentaste.berkeley.edu/dataset/

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
