[Supplementary Material]



Figure 1: (a) Achievable bounds of Algorithm 1 in the adversarial setting for different choices of window length $W$. (b) Excess overall utility of the shifting benchmark sequence with respect to the static benchmark for different numbers of allowed shifts.

## Appendix

## Additional plots

Figure 1(a) demonstrates the achievable adversarial static regret and total constraint violation by Algorithm 1 for different choices of the window length $W$. The black curves indicate the Pareto frontier for different values of $W$ and all the points north-east of the frontier are achievable. In particular, we can observe that for $W = T$, both the regret and total constraint violation of Algorithm 1 are linear.

In Figure 1(b), we have plotted the excess overall utility of the shifting benchmark sequence [1] with respect to the static benchmark action for different number of allowed shifts for experiment 2. In other words, we consider the dynamic benchmark sequence $\{x_t^*\}_{t=1}^T$ which changes at most $m$ times (i.e., $x_t^* \neq x_{t+1}^*$ for at most $m$ values of $t \in \{1, \ldots, T-1\}$) and we plot the excess overall utility for different choices of $m$. Note that the adversarial static and dynamic regret defined in the paper correspond to $m = 0$ and $m = T - 1$ respectively. Figure 1(b) verifies that the dynamic benchmark sequence achieves much higher utility compared to the static benchmark action and therefore, for the settings where the environment is changing, dynamic regret is a more suitable performance metric.

## A    Examples of DR-submodular functions

**Continuous extension of submodular set functions.**    A discrete function $f : \{0, 1\}^V \to \mathbb{R}$ over the ground set $V$ is submodular if for all $j \in V$ and $A \subseteq B \subseteq V \setminus \{j\}$, the following holds:

$$f(A \cup \{j\}) - f(A) \geq f(B \cup \{j\}) - f(B).$$

DR-submodularity is the continuous counterpart of the submodularity property of set functions [2]. Indeed, the multilinear extension [3] and the softmax extension [4] of submodular set functions are DR-submodular.

**Indefinite quadratic functions.**    Let $f(x) = \frac{1}{2}x^T H x + h^T x + c$. If the matrix $H$ is element-wise non-positive, $f$ is a DR-submodular function.

**Concave functions with negative dependence.**    If $h_i : \mathbb{R} \to \mathbb{R}$ is concave for all $i \in [n]$ and $\theta_{ij} \leq 0 \ \forall i \neq j$, the following function $f : \mathbb{R}_+^n \to \mathbb{R}$ is DR-submodular:

$$f(x) = \sum_{i=1}^n h_i(x_i) + \sum_{i,j:i \neq j} \theta_{ij} x_i x_j.$$

Note that indefinite quadratic functions are a special example of the above where all the concave functions $h_i$ are quadratic with negative coefficients.

**Log-determinant function.** Let the function $f : [0, 1]^n \to \mathbb{R}$ be defined as

$$f(x) = \log \det \big(\operatorname{diag}(x)(L - I) + I\big),$$

where $L \succeq 0$ is a positive semidefinite matrix and $\operatorname{diag}(x)$ denotes a diagonal matrix with vector $x$ on its diagonal. This function is extensively used as the utility function in Determinantal Point Processes (DPPs). It was proved in [4] that $f$ is a DR-submodular function. In fact, $f$ is the softmax extension of the submodular set function $\log \det(L_S)$ over the ground set $V$ where $L_S$ is the submatrix of $L$ whose rows and columns are characterized by the set $S \subseteq V$.

## B  Proof of Lemma 1

Denote $r_t^{(k)} = V \nabla f_{t-1}(x_{t-1}^{(k)}) - \lambda_t^{(k)} \nabla g_{t-1}(v_{t-1}^{(k)})$. We have:

$$
\begin{aligned}
v_t^{(k)} &= \arg\max_{x \in \mathcal{X}} \big( \langle r_t^{(k)}, x \rangle - \alpha \| x - v_{t-1}^{(k)} \|^2 \big) \\
&= \arg\min_{x \in \mathcal{X}} \big( - \langle r_t^{(k)}, x \rangle + \alpha \| x - v_{t-1}^{(k)} \|^2 \big) \\
&= \arg\min_{x \in \mathcal{X}} \big( - \langle \frac{r_t^{(k)}}{\alpha}, x \rangle + \| x - v_{t-1}^{(k)} \|^2 \big) \\
&\overset{(a)}{=} \arg\min_{x \in \mathcal{X}} \big( - \langle \frac{r_t^{(k)}}{\alpha}, x - v_{t-1}^{(k)} \rangle + \| x - v_{t-1}^{(k)} \|^2 + \| \frac{r_t^{(k)}}{2\alpha} \|^2 \big) \\
&= \arg\min_{x \in \mathcal{X}} \| x - v_{t-1}^{(k)} - \frac{r_t^{(k)}}{2\alpha} \|^2 \\
&= \mathcal{P}_{\mathcal{X}} \big( v_{t-1}^{(k)} + \frac{r_t^{(k)}}{2\alpha} \big),
\end{aligned}
$$

where in (a), we have added the constant term $\langle \frac{r_t^{(k)}}{\alpha}, v_{t-1}^{(k)} \rangle + \| \frac{r_t^{(k)}}{2\alpha} \|^2$ to complete the square norm which does not affect the minimizer.

## C  Proof of Lemma 2

Using the update rule of the algorithm for $\lambda_{t+1}^{(k)}$, we can write:

$$
\begin{aligned}
\lambda_{t+1}^{(k)} - \lambda_t^{(k)} &\geq g_{t-1}(v_{t-1}^{(k)}) + \langle \nabla g_{t-1}(v_{t-1}^{(k)}), v_t^{(k)} - v_{t-1}^{(k)} \rangle \\
&\geq g_{t-1}(v_{t-1}^{(k)}) - \beta \| v_t^{(k)} - v_{t-1}^{(k)} \| \\
&= g_{t-1}(v_{t-1}^{(k)}) - \frac{\beta^2}{4V} - V \| v_t^{(k)} - v_{t-1}^{(k)} \|^2 + \underbrace{( \frac{\beta}{2\sqrt{V}} - \sqrt{V} \| v_t^{(k)} - v_{t-1}^{(k)} \| )^2}_{\geq 0} \\
&\geq g_{t-1}(v_{t-1}^{(k)}) - \frac{\beta^2}{4V} - V \| v_t^{(k)} - v_{t-1}^{(k)} \|^2,
\end{aligned}
$$

where we have used $\beta$-Lipschitzness of the constraint function $g_{t-1}$ and the Cauchy-Schwarz inequality to obtain the second inequality.
Taking the sum over $t \in [T]$, we obtain:

$$
\lambda_{T+1}^{(k)} - \underbrace{\lambda_1^{(k)}}_{=0} \geq \sum_{t=1}^{T} g_{t-1}(v_{t-1}^{(k)}) - \frac{\beta^2 T}{4V} - V \sum_{t=1}^{T} \| v_t^{(k)} - v_{t-1}^{(k)} \|^2,
$$

$$
\sum_{t=1}^{T} g_{t-1}(v_{t-1}^{(k)}) \leq \lambda_{T+1}^{(k)} + \frac{\beta^2 T}{4V} + V \sum_{t=1}^{T} \| v_t^{(k)} - v_{t-1}^{(k)} \|^2.
$$

Therefore, we have:

$$C_T = \sum_{t=1}^{T} g_{t-1}(x_{t-1})$$

$$= \sum_{t=1}^{T} g_{t-1}\left(\frac{1}{K}\sum_{k=1}^{K} v_{t-1}^{(k)}\right)$$

$$\leq \frac{1}{K}\sum_{k=1}^{K}\sum_{t=1}^{T} g_{t-1}(v_{t-1}^{(k)})$$

$$\leq \frac{1}{K}\sum_{k=1}^{K}\lambda_{T+1}^{(k)} + \frac{\beta^2 T}{4V} + \frac{V}{K}\sum_{k=1}^{K}\sum_{t=1}^{T}\|v_t^{(k)} - v_{t-1}^{(k)}\|^2,$$

where the first inequality is due to Jensen's inequality.

## D  Proof of Lemma 3

Using the update rule of the algorithm for $\lambda_{t+1}^{(k)}$, we have:

$$(\lambda_{t+1}^{(k)})^2 = [\lambda_t^{(k)} + g_{t-1}(v_{t-1}^{(k)}) + \langle \nabla g_{t-1}(v_{t-1}^{(k)}), v_t^{(k)} - v_{t-1}^{(k)}\rangle]_+^2$$

$$\leq \left(\lambda_t^{(k)} + g_{t-1}(v_{t-1}^{(k)}) + \langle \nabla g_{t-1}(v_{t-1}^{(k)}), v_t^{(k)} - v_{t-1}^{(k)}\rangle\right)^2$$

$$= (\lambda_t^{(k)})^2 + (g_{t-1}(v_{t-1}^{(k)}) + \langle \nabla g_{t-1}(v_{t-1}^{(k)}), v_t^{(k)} - v_{t-1}^{(k)}\rangle)^2 + 2\lambda_t^{(k)}(g_{t-1}(v_{t-1}^{(k)}) + \langle \nabla g_{t-1}(v_{t-1}^{(k)}), v_t^{(k)} - v_{t-1}^{(k)}\rangle)$$

$$\leq (\lambda_t^{(k)})^2 + (G + \beta R)^2 + 2\lambda_t^{(k)}(g_{t-1}(v_{t-1}^{(k)}) + \langle \nabla g_{t-1}(v_{t-1}^{(k)}), v_t^{(k)} - v_{t-1}^{(k)}\rangle).$$

So, we can conclude:

$$\Delta_t^{(k)} = \frac{(\lambda_{t+1}^{(k)})^2}{2} - \frac{(\lambda_t^{(k)})^2}{2} \leq \frac{(G + \beta R)^2}{2} + \lambda_t^{(k)}(g_{t-1}(v_{t-1}^{(k)}) + \langle \nabla g_{t-1}(v_{t-1}^{(k)}), v_t^{(k)} - v_{t-1}^{(k)}\rangle).$$

## E  Lemma 4

**Lemma 4.** *The following holds for all $t \in [T]$ and $k \in [K]$:*

$$\left(f_{t-1}(x) - f_{t-1}(x_{t-1}^{(k+1)})\right) \leq \left(1 - \frac{1}{K}\right)\left(f_{t-1}(x) - f_{t-1}(x_{t-1}^{(k)})\right) - \frac{1}{KV}\Delta_t^{(k)} + \frac{V\beta^2}{4\alpha K} + \frac{(G + \beta R)^2}{2KV}$$

$$+ \frac{1}{KV}\lambda_t^{(k)}g_{t-1}(x) + \frac{\alpha}{KV}\|x - v_{t-1}^{(k)}\|^2 - \frac{\alpha}{KV}\|x - v_t^{(k)}\|^2 + \frac{LR^2}{2K^2}. \tag{1}$$

*Proof.* Combining the result of Lemma 3 with the update rule for $v_t^{(k)}$, we have:

$$\Delta_t^{(k)} - \langle V\nabla f_{t-1}(x_{t-1}^{(k)}), v_t^{(k)} - v_{t-1}^{(k)}\rangle + \alpha\|v_t^{(k)} - v_{t-1}^{(k)}\|^2$$

$$\leq \frac{(G + \beta R)^2}{2} - \langle V\nabla f_{t-1}(x_{t-1}^{(k)}) - \lambda_t^{(k)}\nabla g_{t-1}(v_{t-1}^{(k)}), v_t^{(k)} - v_{t-1}^{(k)}\rangle + \lambda_t^{(k)}g_{t-1}(v_{t-1}^{(k)}) + \alpha\|v_t^{(k)} - v_{t-1}^{(k)}\|^2$$

$$\leq \frac{(G + \beta R)^2}{2} - \langle V\nabla f_{t-1}(x_{t-1}^{(k)}) - \lambda_t^{(k)}\nabla g_{t-1}(v_{t-1}^{(k)}), x - v_{t-1}^{(k)}\rangle + \lambda_t^{(k)}g_{t-1}(v_{t-1}^{(k)}) + \alpha\|x - v_{t-1}^{(k)}\|^2 - \alpha\|x - v_t^{(k)}\|^2$$

$$\leq \frac{(G + \beta R)^2}{2} - \langle V\nabla f_{t-1}(x_{t-1}^{(k)}), x - v_{t-1}^{(k)}\rangle + \lambda_t^{(k)}g_{t-1}(x) + \alpha\|x - v_{t-1}^{(k)}\|^2 - \alpha\|x - v_t^{(k)}\|^2, \tag{2}$$

where the second inequality is due to the optimality condition of the optimization problem for updating $v_t^{(k)}$ and the last inequality follows from convexity of the constraint function $g_{t-1}$.
We have:

$$-\langle V\nabla f_{t-1}(x_{t-1}^{(k)}), v_t^{(k)} - v_{t-1}^{(k)}\rangle + \alpha\|v_t^{(k)} - v_{t-1}^{(k)}\|^2 = \|\sqrt{\alpha}(v_t^{(k)} - v_{t-1}^{(k)}) - \frac{V}{2\sqrt{\alpha}}\nabla f_{t-1}(x_{t-1}^{(k)})\|^2 - \frac{V^2}{4\alpha}\|\nabla f_{t-1}(x_{t-1}^{(k)})\|^2$$

$$\geq -\frac{V^2\beta^2}{4\alpha}.$$

Using $L$-smoothness of the utility functions, we can write:
$$f_{t-1}(x_{t-1}^{(k+1)}) \geq f_{t-1}(x_{t-1}^{(k)}) + \frac{1}{K}\langle v_{t-1}^{(k)}, \nabla f_{t-1}(x_{t-1}^{(k)})\rangle - \frac{L}{2K^2}\|v_{t-1}^{(k)}\|^2,$$
$$V\langle v_{t-1}^{(k)}, \nabla f_{t-1}(x_{t-1}^{(k)})\rangle \leq KV\big(f_{t-1}(x_{t-1}^{(k+1)}) - f_{t-1}(x_{t-1}^{(k)})\big) + \frac{LR^2V}{2K}.$$

Using the DR-submodularity and monotonocity of the reward functions, we have:
$$\begin{aligned}
f_{t-1}(x) - f_{t-1}(x_{t-1}^{(k)}) &\leq f_{t-1}(x \vee x_{t-1}^{(k)}) - f_{t-1}(x_{t-1}^{(k)})\\
&\leq \langle \nabla f_{t-1}(x_{t-1}^{(k)}), (x \vee x_{t-1}^{(k)}) - x_{t-1}^{(k)}\rangle\\
&= \langle \nabla f_{t-1}(x_{t-1}^{(k)}), (x - x_{t-1}^{(k)}) \vee 0\rangle\\
&\leq \langle \nabla f_{t-1}(x_{t-1}^{(k)}), x\rangle.
\end{aligned}$$

Putting the above inequalities together, we have:
$$\begin{aligned}
\Delta_t^{(k)} - \frac{V^2\beta^2}{4\alpha} &\leq \frac{(G + \beta R)^2}{2} - V\big(f_{t-1}(x) - f_{t-1}(x_{t-1}^{(k)})\big) + KV\big(f_{t-1}(x_{t-1}^{(k+1)}) - f_{t-1}(x_{t-1}^{(k)})\big)\\
&\quad + \lambda_t^{(k)} g_{t-1}(x) + \alpha\|x - v_{t-1}^{(k)}\|^2 - \alpha\|x - v_t^{(k)}\|^2 + \frac{LR^2V}{2K}.
\end{aligned}$$

Rearranging the terms, we can write:
$$\begin{aligned}
KV\big(f_{t-1}(x) - f_{t-1}(x_{t-1}^{(k+1)})\big) &\leq (K-1)V\big(f_{t-1}(x) - f_{t-1}(x_{t-1}^{(k)})\big) - \Delta_t^{(k)} + \frac{V^2\beta^2}{4\alpha} + \frac{(G + \beta R)^2}{2}\\
&\quad + \lambda_t^{(k)} g_{t-1}(x) + \alpha\|x - v_{t-1}^{(k)}\|^2 - \alpha\|x - v_t^{(k)}\|^2 + \frac{LR^2V}{2K}.
\end{aligned}$$

Dividing both sides by $KV$, we obtain the desired result. $\qquad\square$

## F  Lemma 5

**Lemma 5.** *The static regret of Algorithm 1 against $x \in \mathcal{X}$ is bounded as follows:*
$$\sum_{t=1}^{T}\Big((1-\frac{1}{e})f_{t-1}(x) - f_{t-1}(x_{t-1})\Big) \leq \frac{V\beta^2 T}{4\alpha} + \frac{(G + \beta R)^2 T}{2V} + \frac{1}{KV}\sum_{k=1}^{K}\sum_{t=1}^{T}\lambda_t^{(k)} g_{t-1}(x) + \frac{\alpha R^2}{V} + \frac{LR^2 T}{2K}.$$
$$(3)$$

*Proof.* Using inequality 1 and taking the sum over $t \in [T]$, we obtain:
$$\sum_{t=1}^{T}\big(f_{t-1}(x) - f_{t-1}(x_{t-1}^{(k+1)})\big)$$
$$\leq (1 - \frac{1}{K})\sum_{t=1}^{T}\big(f_{t-1}(x) - f_{t-1}(x_{t-1}^{(k)})\big) - \frac{1}{2KV}(\lambda_{T+1}^{(k)})^2 + \frac{1}{2KV}\underbrace{(\lambda_1^{(k)})^2}_{=0} + \frac{V\beta^2 T}{4\alpha K} + \frac{(G + \beta R)^2 T}{2KV}$$
$$+ \frac{1}{KV}\sum_{t=1}^{T}\lambda_t^{(k)} g_{t-1}(x) + \frac{\alpha}{KV}\|x - v_0^{(k)}\|^2 - \frac{\alpha}{KV}\|x - v_T^{(k)}\|^2 + \frac{LR^2 T}{2K^2}$$
$$\leq (1 - \frac{1}{K})\sum_{t=1}^{T}\big(f_{t-1}(x) - f_{t-1}(x_{t-1}^{(k)})\big) + \frac{V\beta^2 T}{4\alpha K} + \frac{(G + \beta R)^2 T}{2KV} + \frac{1}{KV}\sum_{t=1}^{T}\lambda_t^{(k)} g_{t-1}(x) + \frac{\alpha R^2}{KV} + \frac{LR^2 T}{2K^2}.$$

Applying the above inequality recursively for all $k \in [K]$, we have:
$$\sum_{t=1}^{T}\big(f_{t-1}(x) - f_{t-1}(\underbrace{x_{t-1}^{(K+1)}}_{x_{t-1}})\big) \leq \underbrace{(1 - \frac{1}{K})^K}_{\leq \frac{1}{e}}\sum_{t=1}^{T}\big(f_{t-1}(x) - f_{t-1}(\underbrace{x_{t-1}^{(1)}}_{=0})\big) + \frac{V\beta^2 T}{4\alpha} + \frac{(G + \beta R)^2 T}{2V}$$
$$+ \frac{1}{KV}\sum_{k=1}^{K}\sum_{t=1}^{T}\lambda_t^{(k)} g_{t-1}(x) + \frac{\alpha R^2}{V} + \frac{LR^2 T}{2K}.$$

Rearranging the terms, we obtain the desired result. $\qquad\square$

# G Lemma 6

**Lemma 6.** *For all $t \in [T], k \in [K]$, we have $\lambda_t^{(k)} \leq \theta V$ where*

$$\theta = \max\{G + \beta R, \frac{\frac{(G+\beta R)^2}{2} + (\beta R + \frac{V\beta^2}{4\alpha})V}{VB_T/T} + \frac{\alpha R^2}{V(V+1)B_T/T} + \frac{(G+\beta R)(V+2)}{2V}\}.$$

*In particular, if we choose $\alpha \leq \mathcal{O}(V^2)$, we have $\lambda_t^{(k)} \leq \mathcal{O}(V)$.*

*Proof.* Plugging in $x = 0$ in inequality 2, we obtain:

$$\Delta_t^{(k)} - \frac{V^2\beta^2}{4\alpha} \leq \frac{(G+\beta R)^2}{2} + \underbrace{\langle V\nabla f_{t-1}(x_{t-1}^{(k)}), v_{t-1}^{(k)}\rangle}_{\leq \beta RV} - \frac{B_T}{T}\lambda_t^{(k)} + \alpha\|v_{t-1}^{(k)}\|^2 - \alpha\|v_t^{(k)}\|^2,$$

$$\Delta_t^{(k)} \leq \frac{(G+\beta R)^2}{2} + (\beta R + \frac{V\beta^2}{4\alpha})V - \frac{B_T}{T}\lambda_t^{(k)} + \alpha\|v_{t-1}^{(k)}\|^2 - \alpha\|v_t^{(k)}\|^2.$$

Therefore, setting $\eta = -\frac{B_T}{T}$, $B = \frac{(G+\beta R)^2}{2}$ and $R = \beta R + \frac{V\beta^2}{4\alpha}$ in Thoerem 2 of [5], we obtain the desired result. $\square$

This $\mathcal{O}(V)$ bound on the dual variables is crucial in obtaining improved total constraint violation bounds compared to [6]. Note that [5] requires the extra assumption that there exists an action $z \in \mathcal{X}$ such that $g_t(z) < 0 \; \forall t \in [T]$ (Slater condition) to obtain Theorem 2 in their paper. However, in our framework, since $g_t(\cdot) = h_t(\cdot) - \frac{B_T}{T}$ and $h_t(0) = 0$ for all $t \in [T]$, the Slater condition holds naturally with $z = 0$.

# H Proof of Theorem 1

First, using the result of Lemma 2, we have:

$$C_T \leq \frac{1}{K}\sum_{k=1}^{K}\lambda_{T+1}^{(k)} + \frac{\beta^2 T}{4V} + \frac{V}{K}\sum_{k=1}^{K}\sum_{t=1}^{T}\|v_t^{(k)} - v_{t-1}^{(k)}\|^2.$$

Therefore, in order to bound the total constraint violation, we need to bound $\lambda_{T+1}^{(k)}$ and $\|v_t^{(k)} - v_{t-1}^{(k)}\|^2$ for all $k \in [K]$ and $t \in [T]$. Lemma 6 provides the bound $\lambda_{T+1}^{(k)} \leq \theta V$ for the dual variables. Thus, it suffices to obtain upper bounds for the terms $\|v_t^{(k)} - v_{t-1}^{(k)}\|^2$ which is done in the following. Using the update rule of the algorithm for $v_t^{(k)}$, we have:

$$\langle V\nabla f_{t-1}(x_{t-1}^{(k)}) - \lambda_t^{(k)}\nabla g_{t-1}(v_{t-1}^{(k)}), v_t^{(k)}\rangle - \alpha\|v_t^{(k)} - v_{t-1}^{(k)}\|^2 \geq \langle V\nabla f_{t-1}(x_{t-1}^{(k)}) - \lambda_t^{(k)}\nabla g_{t-1}(v_{t-1}^{(k)}), v_{t-1}^{(k)}\rangle$$
$$+ \alpha\|v_t^{(k)} - v_{t-1}^{(k)}\|^2.$$

Equivalently, we can write:

$$2\alpha\|v_t^{(k)} - v_{t-1}^{(k)}\|^2 \leq \langle V\nabla f_{t-1}(x_{t-1}^{(k)}) - \lambda_t^{(k)}\nabla g_{t-1}(v_{t-1}^{(k)}), v_t^{(k)} - v_{t-1}^{(k)}\rangle$$
$$\leq \|V\nabla f_{t-1}(x_{t-1}^{(k)}) - \lambda_t^{(k)}\nabla g_{t-1}(v_{t-1}^{(k)})\|\|v_t^{(k)} - v_{t-1}^{(k)}\|,$$

where we have used the Cauchy-Schwarz inequality to obtain the last inequality.
Dividing both sides by $2\alpha\|v_t^{(k)} - v_{t-1}^{(k)}\|$ and using the triangle inequality, we obtain:

$$\|v_t^{(k)} - v_{t-1}^{(k)}\| \leq \frac{1}{2\alpha}\left(\|V\nabla f_{t-1}(x_{t-1}^{(k)})\| + \|\lambda_t^{(k)}\nabla g_{t-1}(v_{t-1}^{(k)})\|\right)$$
$$\leq \frac{\beta}{2\alpha}(V + \lambda_t^{(k)})$$
$$\leq \frac{\beta(1+\theta)}{2\alpha}V.$$

Therefore, the following holds:

$$\|v_t^{(k)} - v_{t-1}^{(k)}\|^2 \le \frac{\beta^2(1+\theta)^2}{4\alpha^2}V^2.$$

Plugging the above inequality in the result of Lemma 2, we obtain the total constraint violation bound as follows:

$$C_T \le \frac{1}{K}\sum_{k=1}^{K}\theta V + \frac{\beta^2 T}{4V} + \frac{V}{K}\sum_{k=1}^{K}\sum_{t=1}^{T}\|v_t^{(k)} - v_{t-1}^{(k)}\|^2 \le \theta V + \frac{\beta^2 T}{4V} + \frac{\beta^2(1+\theta)^2 V^3 T}{4\alpha^2}.$$

# I Proof of Theorem 2

First, note that in the adversarial setting with window size $W = 1$, since for all $t \in [T]$ and $k \in [K]$, $g_t(x^*) \le 0$ holds and $\lambda_t^{(k)}$ is non-negative, we have $\frac{1}{KV}\sum_{k=1}^{K}\sum_{t=1}^{T}\lambda_t^{(k)}g_{t-1}(x^*) \le 0$. Thus, in this setting, the result follows immediately from inequality 3.
For the case where $W > 1$, plugging in $t \leftarrow t + \tau$ and $x = x_W^*$ in inequality 1, we have:

$$\big(f_{t+\tau-1}(x_W^*) - f_{t+\tau-1}(x_{t+\tau-1}^{(k+1)})\big) \le \big(1 - \frac{1}{K}\big)\big(f_{t+\tau-1}(x_W^*) - f_{t+\tau-1}(x_{t+\tau-1}^{(k)})\big) - \frac{1}{KV}\Delta_{t+\tau}^{(k)} + \frac{V\beta^2}{4\alpha K}$$
$$+ \frac{(G+\beta R)^2}{2KV} + \frac{LR^2}{2K^2} + \frac{1}{KV}\lambda_{t+\tau}^{(k)}g_{t+\tau-1}(x_W^*) + \frac{\alpha}{KV}\|x_W^* - v_{t+\tau-1}^{(k)}\|^2 - \frac{\alpha}{KV}\|x_W^* - v_{t+\tau}^{(k)}\|^2.$$

Taking the sum over all $t \in [T - W + 1], \tau \in \{0, \dots, W - 1\}$ and applying the inequality recursively for all $k \in [K]$, we obtain:

$$\sum_{t=1}^{T-W+1}\sum_{\tau=0}^{W-1}\big(f_{t+\tau-1}(x_W^*) - f_{t+\tau-1}(\underbrace{x_{t+\tau-1}^{(K+1)}}_{=x_{t+\tau-1}})\big) \le \frac{1}{e}\sum_{t=1}^{T-W+1}\sum_{\tau=0}^{W-1}\big(f_{t+\tau-1}(x_W^*) - f_{t+\tau-1}(\underbrace{x_{t+\tau-1}^{(1)}}_{=0})\big)$$
$$- \frac{1}{KV}\sum_{k=1}^{K}\sum_{\tau=0}^{W-1}\sum_{t=1}^{T-W+1}\Delta_{t+\tau}^{(k)} + \frac{V\beta^2 W(T-W+1)}{4\alpha} + \frac{(G+\beta R)^2 W(T-W+1)}{2V} + \frac{LR^2 W(T-W+1)}{2K}$$
$$+ \frac{1}{KV}\sum_{k=1}^{K}\sum_{t=1}^{T-W+1}\sum_{\tau=0}^{W-1}\lambda_{t+\tau}^{(k)}g_{t+\tau-1}(x_W^*) + \frac{\alpha}{KV}\sum_{k=1}^{K}\sum_{\tau=0}^{W-1}\sum_{t=1}^{T-W+1}\big(\|x_W^* - v_{t+\tau-1}^{(k)}\|^2 - \|x_W^* - v_{t+\tau}^{(k)}\|^2\big).$$

Equivalently, we can write:

$$\underbrace{\sum_{t=1}^{T-W+1}\sum_{\tau=0}^{W-1}\big((1 - \frac{1}{e})f_{t+\tau-1}(x_W^*) - f_{t+\tau-1}(x_{t+\tau-1})\big)}_{(a)} \le \underbrace{-\frac{1}{KV}\sum_{k=1}^{K}\sum_{\tau=0}^{W-1}\sum_{t=1}^{T-W+1}\Delta_{t+\tau}^{(k)}}_{(b)}$$
$$+ \frac{V\beta^2 W(T-W+1)}{4\alpha} + \frac{(G+\beta R)^2 W(T-W+1)}{2V} + \underbrace{\frac{1}{KV}\sum_{k=1}^{K}\sum_{t=1}^{T-W+1}\sum_{\tau=0}^{W-1}\lambda_{t+\tau}^{(k)}g_{t+\tau-1}(x_W^*)}_{(c)}$$
$$+ \underbrace{\frac{\alpha}{KV}\sum_{k=1}^{K}\sum_{\tau=0}^{W-1}\sum_{t=1}^{T-W+1}\big(\|x_W^* - v_{t+\tau-1}^{(k)}\|^2 - \|x_W^* - v_{t+\tau}^{(k)}\|^2\big)}_{(d)} + \frac{LR^2 W(T-W+1)}{2K}. \quad (4)$$

The main challenge in obtaining regret bounds for the $W > 1$ case is to bound terms (a), (b), (c) and (d) in the above inequality. We exploit ideas from the analysis in [6, 7] to obtain these bounds.

We bound term (a) in the following:

$$\text{(a)} = WR_T - \sum_{i=1}^{W-1}(W-i)\Big(\big[(1-\frac{1}{e})f_{i-1}(x_W^*) - f_{i-1}(x_i)\big]$$

$$+ \big[(1-\frac{1}{e})f_{T-i}(x_W^*) - f_{T-i}(x_{T-i})\big]\Big)$$

$$\geq WR_T - 2F\sum_{i=1}^{W-1}(W-i)$$

$$= WR_T - FW(W-1). \tag{5}$$

For the term (b), we have:

$$\text{(b)} = -\frac{1}{2KV}\sum_{k=1}^{K}\sum_{\tau=0}^{W-1}\sum_{t=1}^{T-W+1}\big((\lambda_{t+\tau+1}^{(k)})^2 - (\lambda_{t+\tau}^{(k)})^2\big)$$

$$= -\frac{1}{2KV}\sum_{k=1}^{K}\sum_{\tau=0}^{W-1}\big((\lambda_{T-W+\tau+2}^{(k)})^2 - (\lambda_{\tau+1}^{(k)})^2\big)$$

$$\leq \frac{1}{2KV}\sum_{k=1}^{K}\sum_{\tau=0}^{W-1}(\lambda_{\tau+1}^{(k)})^2$$

$$\leq \frac{1}{2KV}\sum_{k=1}^{K}\sum_{\tau=0}^{W-1}\min\{\theta^2 V^2, \tau^2(G+\beta R)^2\}$$

$$= \frac{1}{2KV}\min\{\theta^2 V^2 KW, (G+\beta R)^2 K\sum_{\tau=0}^{W-1}\tau^2\}$$

$$= \frac{1}{2KV}\min\{\theta^2 V^2 KW, (G+\beta R)^2 K\frac{W(W-1)(2W-1)}{6}\}$$

$$= \frac{1}{2V}\min\{\theta^2 V^2 W, (G+\beta R)^2\frac{W(W-1)(2W-1)}{6}\}, \tag{6}$$

where we have used the dual variable bounds in Lemma 6 and the fact that $\lambda_t^{(k)}$ changes by at most $G + \beta R$ over one slot to obtain the second inequality.

In order to bound (c), we use Lemma 8 of [7] to obtain:

$$\text{(c)} \leq \frac{1}{KV}\sum_{k=1}^{K}\sum_{t=1}^{T-W+1}\Big(\lambda_t^{(k)}\underbrace{\sum_{\tau=0}^{W-1}g_{t+\tau-1}(x_W^*)}_{\leq 0} + \frac{(G+\beta R)^2}{2}W(W-1)\big)\Big)$$

$$\leq \frac{1}{KV}\sum_{k=1}^{K}\sum_{t=1}^{T-W+1}\frac{(G+\beta R)^2}{2}W(W-1)$$

$$= \frac{(G+\beta R)^2 W(W-1)(T-W+1)}{2V}. \tag{7}$$

Finally, for the term (d), we can write:

$$\text{(d)} = \frac{\alpha}{KV}\sum_{k=1}^{K}\sum_{\tau=0}^{W-1}\big(\|x_W^* - v_\tau^{(k)}\|^2 - \|x_W^* - v_{T-W+\tau+1}^{(k)}\|^2\big)$$

$$\leq \frac{\alpha}{KV}\sum_{k=1}^{K}\sum_{\tau=0}^{W-1}\|x_W^* - v_\tau^{(k)}\|^2$$

$$\leq \frac{\alpha}{KV}\sum_{k=1}^{K}\sum_{\tau=0}^{W-1}R^2$$

$$= \frac{\alpha R^2 W}{V}. \tag{8}$$

Combining inequalities 4, 5, 6, 7 and 8, dividing both sides by $W$ and rearranging the terms, we obtain the regret bound as stated.

## J   Proof of Theorem 3

Set $x_0^* = 0$. Plugging in $x = x_{t-1}^*$ in inequality 1, we have:

$$\left(f_{t-1}(x_{t-1}^*) - f_{t-1}(x_{t-1}^{(k+1)})\right) \leq (1 - \frac{1}{K})\left(f_{t-1}(x_{t-1}^*) - f_{t-1}(x_{t-1}^{(k)})\right) - \frac{1}{KV}\Delta_t^{(k)} + \frac{V\beta^2}{4\alpha K} + \frac{(G+\beta R)^2}{2KV}$$

$$+ \frac{1}{KV}\lambda_t^{(k)} g_{t-1}(x_{t-1}^*) + \frac{\alpha}{KV}\|x_{t-1}^* - v_{t-1}^{(k)}\|^2 - \frac{\alpha}{KV}\|x_{t-1}^* - v_t^{(k)}\|^2 + \frac{LR^2}{2K^2}.$$

Taking the sum over all $t \in [T]$ and applying the inequality recursively for all $k \in [K]$, we obtain:

$$\sum_{t=1}^{T}\left(f_{t-1}(x_{t-1}^*) - f_{t-1}(\underbrace{x_{t-1}^{(K+1)}}_{=x_{t-1}})\right) \leq \frac{1}{e}\sum_{t=1}^{T}\left(f_{t-1}(x_{t-1}^*) - f_{t-1}(\underbrace{x_{t-1}^{(1)}}_{=0})\right) + \frac{V\beta^2 T}{4\alpha} + \frac{(G+\beta R)^2 T}{2V}$$

$$+ \frac{1}{KV}\sum_{k=1}^{K}\sum_{t=1}^{T}\lambda_t^{(k)} g_{t-1}(x_{t-1}^*) + \frac{\alpha}{KV}\sum_{k=1}^{K}\sum_{t=1}^{T-1}\left(\|x_t^* - v_t^{(k)}\|^2 - \|x_{t-1}^* - v_t^{(k)}\|^2\right)$$

$$+ \frac{\alpha}{KV}\sum_{k=1}^{K}\underbrace{\|x_0^* - v_0^{(k)}\|^2}_{=0} - \frac{\alpha}{KV}\sum_{k=1}^{K}\|x_{T-1}^* - v_t^{(k)}\|^2 + \frac{LR^2 T}{2K}.$$

Considering that $\|x_t^* - v_t^{(k)}\|^2 - \|x_{t-1}^* - v_t^{(k)}\|^2 = \|x_t^*\|^2 - \|x_{t-1}^*\|^2 + 2\langle v_t^{(k)}, x_{t-1}^* - x_t^*\rangle \leq \|x_t^*\|^2 - \|x_{t-1}^*\|^2 + 2R\|x_{t-1}^* - x_t^*\|$ holds, we can write:

$$\sum_{t=1}^{T}\left(f_{t-1}(x_{t-1}^*) - f_{t-1}(x_{t-1})\right) \leq \frac{1}{e}\sum_{t=1}^{T}f_{t-1}(x_{t-1}^*) + \frac{V\beta^2 T}{4\alpha} + \frac{(G+\beta R)^2 T}{2V} + \frac{1}{KV}\sum_{k=1}^{K}\sum_{t=1}^{T}\lambda_t^{(k)} g_{t-1}(x_{t-1}^*)$$

$$+ \frac{\alpha}{KV}\sum_{k=1}^{K}\left(\underbrace{\|x_{T-1}^*\|^2}_{\leq R^2} - \underbrace{\|x_0^*\|^2}_{=0} + 2R\sum_{t=1}^{T-1}\|x_{t-1}^* - x_t^*\|\right) + \frac{LR^2 T}{2K}.$$

Denoting the drift of the benchmark sequence $P_T^* = \sum_{t=1}^{T-1}\|x_{t-1}^* - x_t^*\|$, we get the dynamic regret bound as desired.

## K   Proof of Theorem 4

Taking expectation of both sides of inequality 1, we have:

$$\mathbb{E}\left[f_{t-1}(x^*) - f_{t-1}(x_{t-1}^{(k+1)})\right] \leq (1 - \frac{1}{K})\mathbb{E}\left[f_{t-1}(x^*) - f_{t-1}(x_{t-1}^{(k)})\right] - \frac{1}{KV}\mathbb{E}[\Delta_t^{(k)}] + \frac{V\beta^2}{4\alpha K} + \frac{(G+\beta R)^2}{2KV}$$

$$+ \frac{1}{KV}\mathbb{E}[\lambda_t^{(k)} g_{t-1}(x^*)] + \frac{\alpha}{KV}\mathbb{E}\|x^* - v_{t-1}^{(k)}\|^2 - \frac{\alpha}{KV}\mathbb{E}\|x^* - v_t^{(k)}\|^2 + \frac{LR^2}{2K^2}.$$

Let $\mathcal{F}_t = \{g_\tau\}_{\tau=0}^{t-1}$. Considering that $\lambda_t^{(k)}$ is $\mathcal{F}_{t-1}$-measurable and $g_{t-1}(x^*)$ is independent of $\mathcal{F}_{t-1}$, we can write:

$$\mathbb{E}[\lambda_t^{(k)} g_{t-1}(x^*)] = \mathbb{E}\left[\mathbb{E}[\lambda_t^{(k)} g_{t-1}(x^*)|\mathcal{F}_{t-1}]\right] = \mathbb{E}\left[\lambda_t^{(k)}\underbrace{\mathbb{E}[g_{t-1}(x^*)]}_{\leq 0}\right] \leq 0.$$

Combining the above inequalities, taking the sum over $t \in [T]$ and applying the inequality recursively for all $k \in [K]$, we obtain:

$$\sum_{t=1}^{T}\mathbb{E}\left[f_{t-1}(x^*) - f_{t-1}(\underbrace{x_{t-1}^{(K+1)}}_{=x_{t-1}})\right] \leq \underbrace{(1 - \frac{1}{K})^K}_{\leq \frac{1}{e}}\mathbb{E}\left[\sum_{t=1}^{T}\left(f_{t-1}(x^*) - f_{t-1}(\underbrace{x_{t-1}^{(1)}}_{=0})\right)\right] \underbrace{-\frac{1}{2KV}\sum_{k=1}^{K}\mathbb{E}[\lambda_{T+1}^{(k)}]^2}_{\leq 0}$$

$$+ \frac{V\beta^2 T}{4\alpha} + \frac{(G+\beta R)^2 T}{2V} + \frac{\alpha R^2}{V} + \frac{LR^2 T}{2K}.$$

Therefore, the expected regret bound is derived as stated.

## L  Proof of Theorem 5

Considering the regret bound in inequality 3, in order to obtain a high probability regret bound, we have to bound $\frac{1}{KV} \sum_{k=1}^{K} \sum_{t=1}^{T} \lambda_t^{(k)} g_{t-1}(x^*)$. Denote $Y_t = \frac{1}{KV} \sum_{k=1}^{K} \sum_{s=1}^{t} \lambda_s^{(k)} g_{s-1}(x^*)$ and let $\mathcal{F}_t = \{g_\tau\}_{\tau=0}^{t-1}$. Considering that $g_{t-1}(x^*)$ is independent of $\mathcal{F}_{t-1}$, We have:

$$\mathbb{E}[Y_t|\mathcal{F}_{t-1}] = \mathbb{E}[Y_{t-1} + \frac{1}{KV} \sum_{k=1}^{K} \lambda_t^{(k)} g_{t-1}(x^*)|\mathcal{F}_{t-1}]$$

$$= Y_{t-1} + \mathbb{E}[\frac{1}{KV} \sum_{k=1}^{K} \lambda_t^{(k)} g_{t-1}(x^*)|\mathcal{F}_{t-1}]$$

$$= Y_{t-1} + \frac{1}{KV} \sum_{k=1}^{K} \lambda_t^{(k)} \mathbb{E}[g_{t-1}(x^*)|\mathcal{F}_{t-1}]$$

$$= Y_{t-1} + \frac{1}{KV} \sum_{k=1}^{K} \lambda_t^{(k)} \underbrace{\mathbb{E}[g_{t-1}(x^*)]}_{\leq 0}$$

$$\leq Y_{t-1}.$$

Therefore, $\{Y_t, \mathcal{F}_t\}_{t\geq 0}$ is a supermartingale. Also, note that for all $t \in [T]$, we have:

$$|Y_t - Y_{t-1}| = |\frac{1}{KV} \sum_{k=1}^{K} \lambda_t^{(k)} g_{t-1}(x^*)| \leq \frac{1}{KV} \sum_{k=1}^{K} \lambda_t^{(k)} |g_{t-1}(x^*)| \leq \theta G.$$

Thus, using the Azuma-Hoeffding inequality, we can conclude that with probability $1 - \delta$, the following holds:

$$\frac{1}{KV} \sum_{k=1}^{K} \sum_{t=1}^{T} \lambda_t^{(k)} g_{t-1}(x^*) \leq \theta G \sqrt{2T\log(\frac{1}{\delta})}.$$

Combining the above inequality with the regret bound in inequality 3, with probability $1 - \delta$, we have:

$$R_T^{(S,S)} \leq \theta G \sqrt{2T\log(\frac{1}{\delta})} + \frac{V\beta^2 T}{4\alpha} + \frac{(G + \beta R)^2 T}{2V} + \frac{\alpha R^2}{V} + \frac{LR^2 T}{2K}.$$

## Algorithm 2

We present our second algorithm for online DR-submodular maximization with adversarial or stochastic constraints in Algorithm 2. Note that in the algorithm, we denote $\bar{g}_q(x) = \frac{\sum_{k=1}^{K} g_{t_{q,k}}(x)}{K} \ \forall q \in [Q], k \in [K]$.

## M  Proof of Theorem 6

Using an analysis similar to Lemma 1, we have:

$$\sum_{q=1}^{Q} \bar{g}_{q-1}(v_{q-1}^{(k)}) \leq \lambda_{Q+1}^{(k)} + \frac{\beta^2 Q}{4V} + V \sum_{q=1}^{Q} \|v_q^{(k)} - v_{q-1}^{(k)}\|^2.$$

---

**Algorithm 2**

---

**Input:** $\mathcal{X}$ is the constraint set, $T$ is the horizon, $V > 0$, $\alpha > 0$ and $K$.
**Output:** $\{x_t : 1 \leq t \leq T\}$.
Initialize $\lambda_1^{(k)} = v_0^{(k)} = x_{t_{0,k}}^{(k)} = 0 \; \forall k \in [K]$.
**for** $q = 1$ **to** $Q$ **do**
  $x_q^{(1)} = 0$.
  **for** $k = 1$ **to** $K$ **do**
    $v_q^{(k)} = \arg\max_{x \in \mathcal{X}} \left( \langle V \nabla f_{t_{q-1,k}}(x_{t_{q-1,k}}^{(k)}) - \lambda_q^{(k)} \nabla \bar{g}_{q-1}(v_{q-1}^{(k)}), x \rangle - \alpha \|x - v_{q-1}^{(k)}\|^2 \right)$,
    $x_q^{(k+1)} = x_q^{(k)} + \frac{1}{K} v_q^{(k)}$.
  **end for**
  Let $(t_{q,1}, \ldots, t_{q,K})$ be a random permutation of $\{(q-1)K+1, \ldots, qK\}$.
  **for** $t = (q-1)K + 1$ **to** $qK$ **do**
    Set $x_t = x_q^{(K+1)}$ and play $x_t$.
  **end for**
  **for** $k = 1$ **to** $K$ **do**
    $\lambda_{q+1}^{(k)} = [\lambda_q^{(k)} + \bar{g}_{q-1}(v_{q-1}^{(k)}) + \langle \nabla \bar{g}_{q-1}(v_{q-1}^{(k)}), v_q^{(k)} - v_{q-1}^{(k)} \rangle]_+$.
  **end for**
**end for**

---

The constraint violation bound follows immediately from the result of Lemma 2 and it is provided below:

$$
\mathbb{E}[C_T] = \mathbb{E}\Big[ K \sum_{q=1}^{Q} \bar{g}_{q-1}(x_{q-1}^{(K+1)}) \Big]
$$

$$
= \mathbb{E}\Big[ K \sum_{q=1}^{Q} \bar{g}_{q-1}\Big( \frac{1}{K} \sum_{k=1}^{K} v_{q-1}^{(k)} \Big) \Big]
$$

$$
\leq \sum_{k=1}^{K} \sum_{q=1}^{Q} \mathbb{E}[\bar{g}_{q-1}(v_{q-1}^{(k)})]
$$

$$
\leq \sum_{k=1}^{K} \mathbb{E}[\lambda_{Q+1}^{(k)}] + \frac{\beta^2 Q K}{4V} + V \sum_{k=1}^{K} \sum_{q=1}^{Q} \mathbb{E}\|v_q^{(k)} - v_{q-1}^{(k)}\|^2, \tag{9}
$$

where the first inequality is due to Jensen's inequality.
Plugging in $t \leftarrow q$ and using the dual update of Algorithm 2 instead of Algorithm 1 in Lemma 3, we have:

$$
\Delta_q^{(k)} := \frac{(\lambda_{q+1}^{(k)})^2}{2} - \frac{(\lambda_q^{(k)})^2}{2} \leq \frac{(G + \beta R)^2}{2} + \lambda_q^{(k)}(\bar{g}_{q-1}(v_{q-1}^{(k)}) + \langle \nabla \bar{g}_{q-1}(v_{q-1}^{(k)}), v_q^{(k)} - v_{q-1}^{(k)} \rangle).
$$

Combining the above inequality with the update rule for $v_q^{(k)}$, we have:

$$
\Delta_q^{(k)} \underbrace{- \langle V \nabla f_{t_{q-1,k}}(x_{t_{q-1,k}}^{(k)}), v_q^{(k)} - v_{q-1}^{(k)} \rangle + \alpha \|v_q^{(k)} - v_{q-1}^{(k)}\|^2}_{(a)}
$$

$$
\leq \frac{(G + \beta R)^2}{2} - \langle V \nabla f_{t_{q-1,k}}(x_{t_{q-1,k}}^{(k)}) - \lambda_q^{(k)} \nabla \bar{g}_{q-1}(v_{q-1}^{(k)}), v_q^{(k)} - v_{q-1}^{(k)} \rangle + \lambda_t^{(k)} \bar{g}_{q-1}(v_{q-1}^{(k)}) + \alpha \|v_q^{(k)} - v_{q-1}^{(k)}\|^2
$$

$$
\leq \frac{(G + \beta R)^2}{2} - \langle V \nabla f_{t_{q-1,k}}(x_{t_{q-1,k}}^{(k)}) - \lambda_q^{(k)} \nabla \bar{g}_{q-1}(v_{q-1}^{(k)}), x - v_{q-1}^{(k)} \rangle + \lambda_q^{(k)} \bar{g}_{q-1}(v_{q-1}^{(k)}) + \alpha \|x - v_{q-1}^{(k)}\|^2 - \alpha \|x - v_q^{(k)}\|^2
$$

$$
\leq \frac{(G + \beta R)^2}{2} - \langle V \nabla f_{t_{q-1,k}}(x_{t_{q-1,k}}^{(k)}), x - v_{q-1}^{(k)} \rangle + \lambda_q^{(k)} \bar{g}_{q-1}(x) + \alpha \|x - v_{q-1}^{(k)}\|^2 - \alpha \|x - v_q^{(k)}\|^2,
$$

where we have used convexity of $\bar{g}_{q-1}$ to derive the last inequality.

For the term (a), we have:

$$(\text{a}) = \|\sqrt{\alpha}(v_q^{(k)} - v_{q-1}^{(k)}) - \frac{V}{2\sqrt{\alpha}}\nabla f_{t_{q-1,k}}(x_{q-1}^{(k)})\|^2 - \frac{V^2}{4\alpha}\|\nabla f_{t_{q-1,k}}(x_{q-1}^{(k)})\|^2$$

$$\geq -\frac{V^2\beta^2}{4\alpha}.$$

Using $L$-smoothness of the utility functions, we can write:

$$f_{t_{q-1,k}}(x_{q-1}^{(k+1)}) \geq f_{t_{q-1,k}}(x_{q-1}^{(k)}) + \frac{1}{K}\langle v_{q-1}^{(k)}, \nabla f_{t_{q-1,k}}(x_{q-1}^{(k)})\rangle - \frac{L}{2K^2}\|v_{q-1}^{(k)}\|^2$$

$$V\langle v_{q-1}^{(k)}, \nabla f_{t_{q-1,k}}(x_{q-1}^{(k)})\rangle \leq KV\big(f_{t_{q-1,k}}(x_{q-1}^{(k+1)}) - f_{t_{q-1,k}}(x_{q-1}^{(k)})\big) + \frac{LR^2V}{2K}.$$

Using the DR-submodularity and monotonocity of the reward functions, we have:

$$f_{t_{q-1,k}}(x) - f_{t_{q-1,k}}(x_{q-1}^{(k)}) \leq f_{t_{q-1,k}}(x \vee x_{q-1}^{(k)}) - f_{t_{q-1,k}}(x_{q-1}^{(k)})$$

$$\leq \langle \nabla f_{t_{q-1,k}}(x_{q-1}^{(k)}), (x \vee x_{q-1}^{(k)}) - x_{q-1}^{(k)}\rangle$$

$$= \langle \nabla f_{t_{q-1,k}}(x_{q-1}^{(k)}), (x - x_{q-1}^{(k)}) \vee 0\rangle$$

$$\leq \langle \nabla f_{t_{q-1,k}}(x_{q-1}^{(k)}), x\rangle.$$

Putting the above inequalities together, we have:

$$\Delta_q^{(k)} - \frac{V^2\beta^2}{4\alpha} \leq \frac{(G+\beta R)^2}{2} - V\big(f_{t_{q-1,k}}(x) - f_{t_{q-1,k}}(x_{q-1}^{(k)})\big) + KV\big(f_{t_{q-1,k}}(x_{q-1}^{(k+1)}) - f_{t_{q-1,k}}(x_{q-1}^{(k)})\big)$$

$$+ \lambda_q^{(k)}\bar{g}_{q-1}(x) + \alpha\|x - v_{q-1}^{(k)}\|^2 - \alpha\|x - v_q^{(k)}\|^2 + \frac{LR^2V}{2K}.$$

Equivalently, we can write:

$$KV\big(f_{t_{q-1,k}}(x) - f_{t_{q-1,k}}(x_{q-1}^{(k+1)})\big) \leq (K-1)V\big(f_{t_{q-1,k}}(x) - f_{t_{q-1,k}}(x_{q-1}^{(k)})\big) - \Delta_q^{(k)} + \frac{V^2\beta^2}{4\alpha} + \frac{(G+\beta R)^2}{2}$$

$$+ \lambda_q^{(k)}\bar{g}_{q-1}(x) + \alpha\|x - v_{q-1}^{(k)}\|^2 - \alpha\|x - v_q^{(k)}\|^2 + \frac{LR^2V}{2K}.$$

Dividing both sides by $KV$ and taking the sum over $q \in [Q]$, we obtain:

$$\sum_{q=1}^{Q}\big(f_{t_{q-1,k}}(x) - f_{t_{q-1,k}}(x_{q-1}^{(k+1)})\big) \leq (1 - \frac{1}{K})\sum_{q=1}^{Q}\big(f_{t_{q-1,k}}(x) - f_{t_{q-1,k}}(x_{q-1}^{(k)})\big) + \frac{V\beta^2 Q}{4\alpha K} + \frac{(G+\beta R)^2 Q}{2KV}$$

$$+ \frac{1}{KV}\sum_{q=1}^{Q}\lambda_q^{(k)}\bar{g}_{q-1}(x) + \frac{\alpha}{KV}\|x - v_0^{(k)}\|^2 - \frac{\alpha}{KV}\|x - v_Q^{(k)}\|^2 + \frac{LR^2 Q}{2K^2}.$$

Applying the above inequality recursively for all $k \in [K]$, we have:

$$\sum_{q=1}^{Q}\big(f_{t_{q-1,k}}(x) - f_{t_{q-1,k}}(x_{q-1}^{(K+1)})\big) \leq \underbrace{(1 - \frac{1}{K})^K}_{\leq \frac{1}{e}}\sum_{q=1}^{Q}\big(f_{t_{q-1,k}}(x) - f_{t_{q-1,k}}(\underbrace{x_{q-1}^{(1)}}_{=0})\big) + \frac{V\beta^2 Q}{4\alpha} + \frac{(G+\beta R)^2 Q}{2V}$$

$$+ \frac{1}{KV}\sum_{k=1}^{K}\sum_{q=1}^{Q}\lambda_q^{(k)}\bar{g}_{q-1}(x) + \frac{\alpha R^2}{V} + \frac{LR^2 Q}{2K}.$$

Therefore, the regret against the benchmark with window length $W \in [1, T^{\frac{1}{3}}]$ is derived as stated.

# N  Proof of Theorem 7

For all $q \in [Q], k \in [K]$, using a similar analysis to Lemma 6, we have $\lambda_q^{(k)} \leq \theta V$ where

$$\theta = \max\{G + \beta R, \frac{\frac{(G+\beta R)^2 T}{2} + (\beta R + \frac{V\beta^2}{4\alpha})VT}{V B_T} + \frac{\alpha R^2 T}{V(V+1)B_T} + \frac{(G+\beta R)(V+2)}{2V}\}.$$

Using the update rule of the algorithm, we have:

$$\langle V\nabla f_{t_{q-1,k}}(x_{q-1}^{(k)}) - \lambda_q^{(k)}\nabla\bar{g}_{q-1}(v_{q-1}^{(k)}), v_q^{(k)}\rangle - \alpha\|v_q^{(k)} - v_{q-1}^{(k)}\|^2 \geq \langle V\nabla f_{t_{q-1,k}}(x_{q-1}^{(k)}) - \lambda_q^{(k)}\nabla\bar{g}_{q-1}(v_{q-1}^{(k)}), v_{q-1}^{(k)}\rangle$$
$$+ \alpha\|v_q^{(k)} - v_{q-1}^{(k)}\|^2.$$

Equivalently, we can write:

$$2\alpha\|v_q^{(k)} - v_{q-1}^{(k)}\|^2 \leq \langle V\nabla f_{t_{q-1,k}}(x_{t_{q-1,k}}^{(k)}) - \lambda_q^{(k)}\nabla\bar{g}_{q-1}(v_{q-1}^{(k)}), v_q^{(k)} - v_{q-1}^{(k)}\rangle$$
$$\leq \|V\nabla f_{t_{q-1,k}}(x_{q-1}^{(k)}) - \lambda_q^{(k)}\nabla\bar{g}_{q-1}(v_{q-1}^{(k)})\|\|v_q^{(k)} - v_{q-1}^{(k)}\|.$$

Dividing both sides by $2\alpha\|v_t^{(k)} - v_{t-1}^{(k)}\|$ and using the triangle inequality, we obtain:

$$\|v_q^{(k)} - v_{q-1}^{(k)}\| \leq \frac{1}{2\alpha}\left(\|V\nabla f_{t_{q-1,k}}(x_{q-1}^{(k)})\| + \|\lambda_q^{(k)}\nabla\bar{g}_{q-1}(v_{q-1}^{(k)})\|\right)$$
$$\leq \frac{\beta}{2\alpha}(V + \lambda_q^{(k)})$$
$$\leq \frac{\beta(1+\theta)}{2\alpha}V.$$

Therefore, the following holds:

$$\|v_q^{(k)} - v_{q-1}^{(k)}\|^2 \leq \frac{\beta^2(1+\theta)^2}{4\alpha^2}V^2.$$

Plugging the above inequality in inequality 9, we obtain the constraint violation bound as follows:

$$\mathbb{E}[C_T] \leq \sum_{k=1}^{K}\theta V + \frac{\beta^2 QK}{4V} + V\sum_{k=1}^{K}\sum_{q=1}^{Q}\mathbb{E}\|v_t^{(k)} - v_{t-1}^{(k)}\|^2 \leq \theta KV + \frac{\beta^2 T}{4V} + \frac{\beta^2(1+\theta)^2 V^3 T}{4\alpha^2}.$$