[Reviews · NeurIPS 2020]

Review 1

Summary and Contributions: After reading the authors' response, I remain positive about the paper. === This paper studies online DR-submodular maximization problemn with convex constraints. The authors proposed an algorithm that achieves sublinear regret and sublinear constraint violation under both stochastic and adversarial models.

Strengths: The authors proposed an unified approach that enjoys theoretical guarantees in both adversarial and stochastic settings.

Weaknesses: This paper may lack discussion on lower bounds for the problem in convex and DR-submodular setting. The authors are recommended to discuss the optimality/suboptimality of the previous works and the proposed algorithm. Fig 1 has several issues. 1) The fonts look very tiny and readers have to zoom in. 2) It would be better to give names to Alg 1 and 2 so that future works can refer to them conveniently. 3) I see a red vertical line (Alg 2) in fig1b. It looks a little bizarre. 4) Is there any reason that can account for the decline of Alg 1 in fig1b starting from t=0.3? For the experiment section, the authors may want to consider comparing the proposed algorithm to Meta-Frank-Wolfe, although it is not designed for a problem with a violation constraint.

Correctness: I didn't check the proofs in the appendix and the main body looks correct to me.

Clarity: Yes

Relation to Prior Work: Yes

Reproducibility: Yes

Additional Feedback:


Review 2

Summary and Contributions: In this paper, the author(s) studied the online DR-submodular maximization problem, under adversarial and stochastic settings, with convex constraints. Two algorithms are proposed, and both sublinear regret bound and sublinear total constraint violation bound are established.

Strengths: This work is believed to be the first work which achieves sublinear bound for cumulative constraint under a general convex long-term constraints. The regret bound is established in terms of both expectation and high probability. In order to obtain more efficiency, a faster algorithms is proposed, too. There are also plenty of empirical experiments (e.g., online joke recommendation, online task assignment in crowdsourcing markets, online welfare maximization with production cost), which validate the performance of the proposed algorithms.

Weaknesses: It seems that the proposed algorithms are straightforward combination of existing online submodular maximisation algorithms ([18, 20]) and the algorithm for online convex problem with adversarial cumulative constraints ([23]). So the novelty of this work should be highlighted. For example, what is the difficulty when we try to combine these two methods, and how the author(s) circumvented the obstacles. Please also emphasize the new ideas/techniques utilized in the work.

Correctness: The claims and results look correct to me, although I didn’t check the proof carefully.

Clarity: This paper is well-written and easy to follow. One suggestion: in line 172, the definition of x_t^* should be provided.

Relation to Prior Work: The motivation and contributions of this work (compared with the existing works) are well explained and highlighted.

Reproducibility: Yes

Additional Feedback: My major concern is the novelty of this work. The ideas and techniques are not sufficiently original to be accepted by NeurIPS. ------ Update: The author(s) addressed my concerns about the novelty in the response. So I raised the score to "marginally above the acceptance threshold".


Review 3

Summary and Contributions: The paper considers the problem of maximizing a general monotone DR-submodular function subject to a general convex constraint (general up to some natural assumptions) in the online regret-minimization setting. The paper presents two algorithms for this problem, and proves bounds on their regret (with respect to the 1-1/e offline approximation) as well as the extent to which they violate the constraint on average. In that respect, the paper considers three kinds of regrets: - The traditional adversarial static regret in which the input is selected by an adversary and the algorithm competes with the best single solution in hindsight. - An adversarial dynamic regret in which the input is selected by an adversary and the algorithm competes with the best offline algorithm. - A stochastic static regret in which the constraint in every round is chosen i.i.d. from some unknown distribution, and the algorithm competes with the best single solution that obeys the expected constraint. For some of these benchmarks there are previous results for the special case in which the constraints are linear. The current paper improves over them both in terms of the generality of the constraint, and in terms of the quality of the guarantees.

Strengths: The paper considers a very general and natural model, and improves in this model over the previous state-of-the-art. The paper also presents a single algorithm that handles both adversarial and stochastic regrets. However, I believe that the last point is less impressive than what one might expect due to the following reason. Algorithms in submodular maximization tend to be quite natural and simple, and the sophistication is mostly in their analyses. Since the algorithm of the paper is of that kind, the fact that the same natural strategy works with respect to different benchmarks (with different analyses) is not very surprising.

Weaknesses: The paper seems to be a bit incremental in terms of the techniques. The algorithms presented are quite similar to previous algorithms for related problems. There might be more novelty in their analysis, but this is not discussed at all in the main part of the paper.

Correctness: Unfortunately, I am unable to assess the correctness of the paper. All the proofs have been deferred to the additional material, which I have only skimmed. However, the results look plausible.

Clarity: Most the paper is well written. In particular, the paper tries to convey in a nice way the intuition behind its algorithms despite the fact that naturally the proofs are deferred to the additional material. The exception is the experiments section, which is very dense, and almost decipherable without first reading about the studied application in the additional material.

Relation to Prior Work: Yes, the previous work is extensively discussed.

Reproducibility: Yes

Additional Feedback: - The response was read, and answers my questions regarding the horizon and the choice of the objective functions. Thus, I keep my (positive) score. - I did not like the deferring of the notation to the appendix. This makes it basically impossible to read the main paper without the appendix, which contradicts the original reason for the separation between the two. - One should explain already in the introduction that the horizon T is known beforehand. This might be well-known for people who are familiar with the particular previous works to which the current paper relates, but it is not obvious for others. - More information about the experiments should appear in the main part of the paper. In particular, I would like to see an explanation for the choice of the objective functions used.


Review 4

Summary and Contributions: This paper investigates the solution of online optimization problems where the objective to maximize is DR-submodular and (a set of) convex averaged constraints must be satisfied. The paper proposes a novel algorithm, provides theoretical analysis and presents experimental results.

Strengths: The paper is extremely well written an easy to follow. The problem is relevant both theoretically and practically. The problem is timely and topically. Generalizes and unifies previous existing results.

Weaknesses: I am happy with the paper as it is. The format of some equations could be enhanced (size of parenthesis, inline fractions...), but this is up to the authors.

Correctness: Yes, they are.

Clarity: Extremely clear.

Relation to Prior Work: First round: Yes, it is. Additional comments/works in the context of stochastic dual optimization could have been included, but this is a minor issue. Second round: As I said in the first round, I think that the paper is a solid contribution. I understand that there are some similarities with respect to previous works (related algorithms) but the theoretical analysis here is stronger, so that I do not see a problem there. BTW, it is not clear if my suggestion will be taken into account (for sure, the authors' response does not explicitly mention it).

Reproducibility: Yes

Additional Feedback: If possible change the format of some of the equations, write a couple of comments on the relation to stochastic dual optimization and expand the numerical results (in the supplementary material, suggested, not required).

[Author Response · NeurIPS 2020]

Thanks to all the reviewers for their constructive feedback, we respond to the major points below (other comments and
suggestions will be applied to the final version of the paper).
• Lower bounds for the regret and $C_T$. To the best of our knowledge, There are no existing minimax optimal bounds
(simultaneously optimal for both the regret and $C_T$) available even for the convex case (the bounds for the algorithms in
prior works are not proven to be minimax optimal either). For the static regret, the $\mathcal{O}(\sqrt{T})$ lower bound on the regret
from the Online Convex Optimization (OCO) literature applies here as well. However, for $C_T$, due to the trade-off
between the regret and $C_T$, it is impossible to obtain lower bounds because obtaining better $C_T$ bounds is always
possible through incurring higher regret. For instance, one can always choose the action $0$ which leads to $C_T < 0$. In
the adversarial setting with $W = 1$, $\mathcal{O}(\sqrt{T})$ bounds for $R_{1,T}^{(A,S)}$ and $C_T$ is the best so far in prior works in the convex
setting and Algorithm 1 achieves the same bounds in the (non-convex) submodular framework.
• Issues with the plots. We will increase the font size of the plots, and give names to the algorithms in the final version
of the paper. The decline in the plot for Algorithm 1 is due to the fact that once the dual variables get large enough
$(\Theta(V))$, $V\nabla f_{t-1}(x_{t-1}^{(k)})$ and $\lambda_t^{(k)}\nabla g_{t-1}(v_{t-1}^{(k)})$ in the update rule of $v_t^{(k)}$ are of the same order, the algorithm becomes
less aggressive in terms of utility maximization and it tries to balance its budget consumption and overall utility. This
decline verifies our improved theoretical guarantee for $C_T$ (compared to [16] and [17]) as well.
• Compare the algorithms with Meta-Frank-Wolfe. The primal update of the OSPHG and OLFW algorithms is the
Meta-Frank-Wolfe algorithm for submodular maximization applied to the Lagrangian function and these two algorithms
take into account the budget consumption as well. So, Meta-Frank-Wolfe algorithm can be viewed as a special case of
these two algorithms with dual variable being set to zero, and it obtains a higher overall utility at the expense of further
violating the budget constraint.
• Obstacles/challenges of applying prior works to our framework and the novelties/ideas in this work.
**Limitations of the OSPHG algorithm.** For the adversarial setting, the OSPHG algorithm obtains a $R_{W,T}^{(A,S)}$ bound of
$O(\sqrt{WT})$ and a $C_T$ bound of $O(W^{1/4}T^{3/4})$ that could be adapted to obtain expected $\mathcal{O}(\sqrt{T})$ and $\mathcal{O}(T^{3/4})$ bounds
for $R_T^{(S,S)}$ and $C_T$ respectively in the stochastic setting. However, in order to obtain *better* $C_T$ bounds in both settings
and obtain *any* high probability bounds in the stochastic setting, a different approach is needed.
**Limitations of the OLFW algorithm.** For the OLFW algorithm, the expected regret bound is sub-optimal (In our
paper, our dual update is such that $\lambda_t^{(k)}$ is $\mathcal{F}_{t-1}$-measurable and $g_{t-1}(x^*)$ is independent of $\mathcal{F}_{t-1}$, where $\mathcal{F}_t = \{g_\tau\}_{\tau=0}^{t-1}$,
which makes it possible to conclude $\mathbb{E}[\lambda_t^{(k)}g_{t-1}(x^*)] = \mathbb{E}\big[\lambda_t^{(k)}\mathbb{E}[g_{t-1}(x^*)]\big] \leq 0$, whereas this term is the dominating
$\mathcal{O}(T^{3/4})$ term in the regret analysis of the OLFW algorithm). Moreover, there are no performance guarantees in the
adversarial setting. In fact, the update rule for the dual variable in the OLFW algorithm is only reasonable when a good
estimate of the constraint functions are available which is not the case in the adversarial setting.
**How to deal with convex constraints?** Neither of OSPHG and OLFW algorithms are able to deal with online convex
constraints. Both these algorithms apply the Meta-Frank-Wolfe algorithm to the Lagrangian as the update rule for the
primal variable. Although this approach makes it possible to get sub-linear $(1 - \frac{1}{e})$-regret bounds, it is not the ideal
way to treat the constraints and that is why they need to further restrict the constraint functions to be linear (i.e., $\nabla g_t$
being fixed) to obtain performance bounds. In order to remedy this issue, we treat the utility and the constraint in the
Lagrangian function differently and we use $v_{t-1}^{(k)}$ (as opposed to $x_{t-1}^{(k)}$) as the argument for $\nabla g_{t-1}$ in the update rule
$v_t^{(k)} = \mathcal{P}_\mathcal{X}\left(v_{t-1}^{(k)} + \frac{1}{2\alpha}\big(V\nabla f_{t-1}(x_{t-1}^{(k)}) - \lambda_t^{(k)}\nabla g_{t-1}(v_{t-1}^{(k)})\big)\right)$ which makes it possible to deal with convex constraints.
**K dual variables needed.** Using a single dual variable (which is done in [15], [16], [17] and [23]) is not enough to
obtain the $\mathcal{O}(\sqrt{T})$ bounds for $C_T$ simultaneously in both adversarial and stochastic settings and we instead maintain $K$
dual variables which further complicates the analysis.
**Comparison with [15] and [23].** Compared to [15], they only obtain regret and $C_T$ bounds in the adversarial setting
for the convex problem (no stochastic analysis). Also, compared to [23], their adversarial analysis for the convex
problem is only done for the special case with window length $W = 1$ and in the stochastic setting, they only obtain
bounds in expectation and they do not provide high probability performance guarantees (while Theorem 5 in our work
provides high probability bounds). In summary, through our proposed algorithms and their analysis, we manage to
address all the limitations of each of the prior works while maintaining their strengths and we provide a unified approach
for all online submodular maximization problems with online convex constraints.
• Define $x_t^*$ in line 172. $x_t^*$ is any arbitrary action in the domain which satisfies the corresponding constraint $g_t(x_t^*) \leq 0$
and it does not need to be the instantaneous maximizer at round $t$. We will make this point clearer.
• Mention that $T$ is known in advance. We are assuming that the horizon $T$ is known in advance and the parameters
of the proposed algorithms are in terms of $T$. However, had we not known $T$ in advance, we could have used the
well-known doubling trick to obtain the same regret and $C_T$ bounds with slightly worse constants. We will specify this
point in the final version of the paper.
• Explanations for the choice of DR-submodular functions as the objectives in the experiments. More explanations and
motivations for the choice of functions are provided in Appendix B (and will be added to the final version of the paper).

[Meta-Review · NeurIPS 2020]

This was a unanimous accept.